# RobGrad: Robustness-Driven Gradient Descent for Stochastic Multi-objective Optimization

## Abstract

Stochastic multi-objective optimization (SMOO) has become an influential framework for many machine learning problems with multiple objectives, where the gradient conflict problem is a fundamental bottleneck for effective training of models. Most existing methods address this problem with gradient-based approaches, which find an optimization direction that improves each objective through gradient manipulation techniques. However, these methods are based on instantaneous gradients and lack a global optimization perspective, which may lead to suboptimal solutions. In this paper, we consider minimizing the worst-case objective value from a global optimization perspective and transform the SMOO problem into a min-max optimization problem. Further, theoretical correspondences between this min-max problem and the SMOO problem are established. Based on this, we propose a robustness-driven gradient descent (RobGrad) algorithm. RobGrad guarantees that each objective performs not badly from a global perspective without introducing additional a priori parameters. Furthermore, we establish non-asymptotic convergence upper bounds for RobGrad in both convex and non-convex settings, which portray the expected performance gap under the worst weight assignment and the rate of RobGrad's decision approaching a Pareto stationary solution. Extensive experiments show that RobGrad has competitive or improved performance compared to state-of-the-art SMOO methods in a series of tasks on multi-task learning. Code is available at https://anonymous.4open.science/r/RobGrad.

## 1 Introduction

In real applications, many machine learning problems face the challenge of stochastic multi-objective optimization (SMOO) (Liu & Vicente, 2022; 2024). These problems essentially require the simultaneous optimization of conflicting objective functions in a stochastic environment, where the stochasticity originates from noise in data collection and sampling, etc. Conflicting objectives usually lead to opposite optimization directions, while stochasticity introduces uncertainty in the optimization process, collectively complicating model training (Zhou et al., 2022).

Existing SMOO methods are broadly classified into two categories (Abdelaziz, 2012; Liu & Vicente, 2024): the *stochastic* approaches and the *multi-objective* approaches. The *stochastic* approaches are to transform SMOO problems into single-objective stochastic optimization problems by scalarization methods (Xin et al., 2022; Kurin et al., 2022; Mahdavi et al., 2013; Lyu et al., 2024). Scalarization methods reformulate the SMOO problem from global perspectives, establishing well-defined theoretical correspondences between scalarized solutions and their projections on the Pareto front of the SMOO problems. However, these methods often require prior knowledge to determine the critical parameters for problem transformation. This is often difficult to determine in practice and brings an additional burden to users.

Commonly used *multi-objective* approaches are sample average approximation methods (Fliege & Xu, 2011; Zhao et al., 2021; 2024), which transform an SMOO problem into a deterministic multi-objective optimization problem through empirical expectation estimates. However, the effectiveness of these methods often depends on the statistical representativeness of the sample distribution and

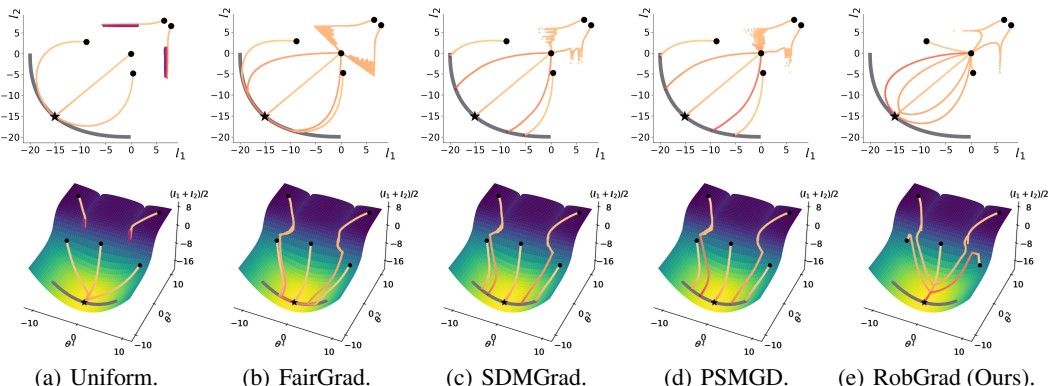

(a) Uniform.  (b) FairGrad.  (c) SDMGrad.  (d) PSMGD.  (e) RobGrad (Ours).

Figure 1: Comparison of SMOO algorithms on a synthetic two-objective problem (Liu et al., 2021a). The optimization trajectories of algorithms based on five different initial points in the objective space $(l_1, l_2$, top row) and the decision-mean objective space $((\theta^1, \theta^2)$-$(l_1 + l_2)/2$, bottom row) are plotted. In figures, the black dots ($\bullet$) are initial points, the gray curve is the Pareto front, and the black pentagram ($\bigstar$) is the Pareto front point with balanced performance. Compared to Uniform and FairGrad in Figures 1(a) and 1(b), RobGrad escapes the local optimum more quickly. Compared to SDMGrad and PSMGD in Figures 1(c) and 1(d), RobGrad is less sensitive to initial points and can stably converge to the Pareto front point with balanced performance.

their theoretical guarantees rely on asymptotic convergence assumptions, which often fail to hold in practical finite-sample scenarios.

Recently, gradient-based *multi-objective* approaches have been widely studied (Mercier et al., 2018; Zhou et al., 2022; Xiao et al., 2023; Fernando et al., 2023; Liu & Vicente, 2024; Xu et al., 2025). These methods find an optimization direction that improves each objective based on multiple instantaneous gradients and simply use a gradient descent step to update decisions. These methods do not require additional prior knowledge, but their myopic reliance on instantaneous gradient information introduces the limitation of lacking a global optimization perspective. The optimization result is sensitive to the initial decision and the resulting vector of objective values may even be close to a minimal point on the Pareto front, meaning that at least one of the other objectives is not good. This is unsatisfactory in most cases, because we tend to make each objective not too bad. This exposes a critical research gap:

*Can we develop a gradient-based SMOO method to overcome the above mentioned drawbacks and achieve the following goals?*

- The method is designed from a global perspective without additional a priori parameters and has non-asymptotic convergence guarantees.

- The method is insensitive to initial decisions and converges stably to the Pareto front point with performance balance.

In this paper, we give a positive answer to the above question. To resolve potential conflicts among objectives in an SMOO problem from a global optimization perspective, we consider minimizing the worst-case objective value and reformulate the SMOO problem as a min-max problem. Based on this, we propose a **Rob**ustness-driven **Grad**ient descent (RobGrad) algorithm to obtain a Pareto front point that makes each objective not bad. Specifically, we make the following major contributions:

- We reformulate an SMOO problem as a min-max problem and establish correspondences between this min-max problem and the SMOO problem. On this basis, we propose the RobGrad algorithm, which is simple and robust for the worst case of weight assignment. Figure 1 shows the practical advantages of RobGrad, which stably obtains the Pareto front point with balanced performance.

- For the convex setting, we derive the non-asymptotic rate convergence of the expected performance gap of RobGrad under the worst weight assignment for the min-max optimization

problem. For non-convex objectives, we further establish a non-asymptotic convergence upper bound of RobGrad for Pareto stationary points.

- We conduct extensive experiments in multi-task learning on multiple datasets to demonstrate the effectiveness of RobGrad, which shows comparable or even better performance compared to existing SMOO algorithms.

## 2 RELATED WORKS

In this section, we briefly introduce the SMOO methods of the *stochastic* and the *multi-objective* approaches. In the *stochastic* approaches, the weighted sum methods (Xin et al., 2022; Kurin et al., 2022) transformed the SMOO problem into a single-objective problem by using pre-defined weights, which belongs to a simplex. By use the $\epsilon$-constraint method in multi-objective optimization (Ehrgott, 2005), Mahdavi et al. (2013) chose one objective as the main objective and bound others by a set of pre-defined $\epsilon$-thresholds. Lyu et al. (2024) transformed the SMOO problem into a quadratic programming problem based on a predefined ideal point, which consists of minimal values for each objective. The theoretical correspondences between the transformed problem and the SMOO problem in these methods are well established, which facilitates the stability of the optimization results. However, the additional prior parameters (e.g., weight vectors, $\epsilon$-thresholds or the ideal point) introduced by these methods pose challenges in practice. In the *multi-objective* approaches, Fliege & Xu (2011); Zhao et al. (2021) used stochastic average approximation methods with a fixed sample size. Zhao et al. (2024) introduced an adaptive sampling mechanism that balances computational efficiency and accuracy. However, these methods require large samples to guarantee convergence.

The gradient-based SMOO method originates from a class of multi-objective gradient manipulation techniques (Fliege & Svaiter, 2000; Désidéri, 2012). They are based on the first-order necessary condition for Pareto optimality in nature, where Multiple-Gradient Descent Algorithm (MGDA) (Désidéri, 2012) is a foundational algorithm in this regard. Sener & Koltun (2018) applied the MGDA to multi-task learning. Yu et al. (2020) further considered the conflict among multiple gradients and proposed PCGrad based on a projection strategy. Liu et al. (2021a) proposed CAGrad, which finds the optimal direction in the sphere centred on the mean gradient to reduce the conflict. However, these algorithms are designed for deterministic situations. Zhou et al. (2022) analyzed the non-convergence of the MGDA, PCGrad and CAGrad methods in stochastic cases. Moreover, stochastic variants of MGDA were introduced (Mercier et al., 2018; Fernando et al., 2023; Liu & Vicente, 2024). Xiao et al. (2023) proposed an SMOO method called SDMGrad to find a gradient direction for optimization by maximizing the minimum decrease across all objectives. Xu et al. (2025) proposed a periodic stochastic multi-gradient descent (PSMGD) to calculate dynamic weights infrequently. However, these methods have myopic dependence on the instantaneous gradient, which leads to the sensitivity of their optimization results to initial decisions.

## 3 PRELIMINARIES

In this section, we first present some notation and basic concepts of SMOO. Then, we illustrate the role of global perspectives in optimization through a pedagogical example.

### 3.1 STOCHASTIC MULTI-OBJECTIVE OPTIMIZATION

Throughout the paper vectors are column vectors written in boldface. For a vector $\boldsymbol{x} \in \mathbb{R}^p$, we write $\boldsymbol{x}^i$ to denote the $i$-th component of a vector $\boldsymbol{x}$. For $p \in \mathbb{N}$, we denote by $[p]$ the set $\{1, 2, \cdots, p\}$. We write $\mathbb{R}_{\geqslant}^p := \{\boldsymbol{x} \in \mathbb{R}^p \mid \forall i \in [p], \boldsymbol{x}^i \geqslant 0\}$ to denote the $p$-dimensional non-negative orthant and denote $\mathbb{R}_{>}^p := \{\boldsymbol{x} \in \mathbb{R}^p \mid \forall i \in [p], \boldsymbol{x}^i > 0\}$ as the $p$-dimensional positive orthant. We denote by $\boldsymbol{e} \in \mathbb{R}^p$ as a vector in which each entry is 1. We denote the $(p-1)$)-dimensional simplex as $\Lambda := \{\boldsymbol{\lambda} \in \mathbb{R}^p \mid \boldsymbol{\lambda}^i \geqslant 0, \boldsymbol{e}^T \boldsymbol{\lambda} = 1\}$ and the interior of a set $\mathcal{S}$ as $\text{int}\mathcal{S}$.

An SMOO problem with $p \geqslant 2$ objectives (Xiao et al., 2023) can be mathematically formulated as:

$$\min_{\boldsymbol{x} \in \mathbb{R}^n} \boldsymbol{f}(\boldsymbol{x}) = (\mathbb{E}_{\boldsymbol{\xi}}[f^1(\boldsymbol{x}, \boldsymbol{\xi})], \cdots, \mathbb{E}_{\boldsymbol{\xi}}[f^p(\boldsymbol{x}, \boldsymbol{\xi})])^T, \quad (1)$$

where $\boldsymbol{x} \in \mathbb{R}^n$ are decision variables and $\boldsymbol{\xi}$ are stochastic variables. The image of the feasible set $\mathbb{R}^n$ under the objective function $\boldsymbol{f}(\cdot)$ is denoted as $Y := \boldsymbol{f}(\mathbb{R}^n) \subseteq \mathbb{R}^p$. Unlike single-objective

optimization problems, an SMOO problem does not have a unique optimal solution, but a set of solutions that do not dominate each other (Ehrgott, 2005; Caramia et al., 2020). The images of these solutions in the objective space are called the Pareto front, which is defined formally as follows.

**Definition 1** ((Weakly) Non-dominance). *For the feasible set $Y$ in the objective space, the non-dominated set and the weakly non-dominated set of $Y$ are defined as:*

$$Y_N := \left\{ \boldsymbol{y} \in Y : (\{\boldsymbol{y}\} - \mathbb{R}^p_{\geqq})\backslash\{0\} \cap Y = \emptyset \right\}, \quad Y_{WN} := \left\{ \boldsymbol{y} \in Y : (\{\boldsymbol{y}\} - int\mathbb{R}^p_{\geqq}) \cap Y = \emptyset \right\},$$

*where the non-dominated set are also called Pareto front.*

**Definition 2** ((Weakly) Efficient Solution). *For the SMOO problem (1), a solution $\boldsymbol{x} \in \mathbb{R}^n$ is called a (weakly) efficient solution if ($\boldsymbol{f}(\boldsymbol{x}) \in Y_{WN}$) $\boldsymbol{f}(\boldsymbol{x}) \in Y_N$.*

Obviously, according to Definitions 1 and 2, we have $Y_N \subseteq Y_{WN}$ and the set of efficient solutions is a subset of the set of weakly efficient solutions.

**Definition 3** (Pareto Stationarity). *For SMOO problem (1), a solution $\boldsymbol{x} \in \mathbb{R}^n$ is called Pareto stationary if $\min_{\boldsymbol{\lambda} \in \Lambda} \|\nabla \boldsymbol{f}(\boldsymbol{x})^T \boldsymbol{\lambda}\| = 0$.*

**Definition 4** ($L$-smoothness). *A function $f : X \to Y$ is $L$-smooth ($L \geqslant 0$) if its gradient is Lipschitz continuous over the convex set $X$, i.e., $\|\nabla f(\boldsymbol{x}) - \nabla f(\boldsymbol{x}')\| \leqslant L\|\boldsymbol{x} - \boldsymbol{x}'\|$ for any $\boldsymbol{x}, \boldsymbol{x}' \in \mathbb{R}^n$.*

## 3.2 A PEDAGOGICAL EXAMPLE

We consider a linear regression problem that has well-studied convex objectives. In machine learning, for a linear regression problem, the objective function usually combines an empirical loss $l(\boldsymbol{x})$ and a regularization term $r(\boldsymbol{x})$, which often conflict. Specifically, minimizing empirical loss $l(\boldsymbol{x})$ may lead to overfitting, while strong regularization $r(\boldsymbol{x})$ may lead to underfitting of the data. Moreover, mini-batch training introduces stochasticity. Thus, a linear regression problem can be formulated as an SMOO problem $P_e : \min_{\boldsymbol{x}} \boldsymbol{f}(\boldsymbol{x}) = (\mathbb{E}_{\boldsymbol{\xi}}[l(\boldsymbol{x}, \boldsymbol{\xi})], \mathbb{E}_{\boldsymbol{\xi}}[r(\boldsymbol{x}, \boldsymbol{\xi})])^T$, treating empirical loss and regularization as two separate objectives that need to be balanced.

In this pedagogical example, we use an online benchmark dataset **mg** from the UCI Machine Learning repository (Dua et al., 2017). We run three algorithms on this example: the weighted sum method with equal weights (representative of scalarization methods), MGDA (representative of gradient-based methods) and RobGrad. The weighted sum method transforms problem $P_e$ into the problem $P_{ws} : \min_{\boldsymbol{x}} 0.5 * l(\boldsymbol{x}) + 0.5 * r(\boldsymbol{x})$. Obviously, problem $P_{ws}$ is a convex problem. Since the weights $\boldsymbol{\lambda} = (0.5, 0.5)^T \in \mathbb{R}^p_{>}$, the optimal solution $\boldsymbol{x}^*_{ws}$ of $P_{ws}$ is an efficient solution of the SMOO problem $P_e$ (Ehrgott, 2005). Then, according to the first-order condition, we can obtain the global optimal solution $\boldsymbol{x}^*_{ws}$, which satisfies $\frac{dr(\boldsymbol{x}^*_{ws})}{dl(\boldsymbol{x}^*_{ws})} = -1$. Thus, from a global perspective, the results of the weighted sum method theoretically converge stably to the Pareto front point with a slope of $-1$.

MGDA obtains the weights $\boldsymbol{\lambda}$ of the joint gradient at each round $k$ by solving a quadratic programming problem, i.e., $\boldsymbol{\lambda}_k = arg\min_{\boldsymbol{\lambda} \in \Lambda} \|\nabla \boldsymbol{f}(\boldsymbol{x}_k, \boldsymbol{\xi})^T \boldsymbol{\lambda}\|^2$. Then, it updates the decision $\boldsymbol{x}$ with a descent step, i.e., $\boldsymbol{x}_{k+1} = \boldsymbol{x}_k - \eta_k \nabla \boldsymbol{f}(\boldsymbol{x}_k)^T \boldsymbol{\lambda}_k$. Thus, MGDA determines the optimization direction in each round based only on the instantaneous gradients, and the results of its optimization depend on the initial solution setup. We set up three different initial points in the experiment and plot the results of these three algorithms on the mg dataset in Figure 2. The results show that for three different initial points, the weighted sum method consistently converges to the point of the Pareto front with slope $-1$. MGDA obtains three different Pareto front points for the three initial solutions, indicating its high sensitivity to initial solutions. Our RobGrad can stably converge

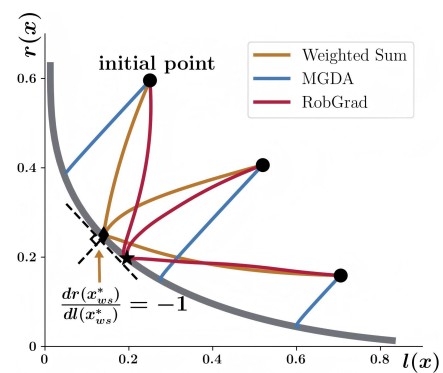

Figure 2: The results on the pedagogical example.

to the performance-balanced Pareto front point for different initial points. These results are consistent with the above analysis, i.e., the optimization results of algorithms with global perspectives are more stable than gradient-based algorithms that only myopically rely on instantaneous gradients.

---

**Algorithm 1** Robustness-Driven Gradient descent (RobGrad)

**Input:** $X$, $K$, $\{\eta_k\}$, $\{\alpha_k\}$, $\{\beta_k\}$, $\boldsymbol{x}_1 \in \mathbb{R}^n$, $\boldsymbol{\lambda}_1 \in \Lambda$.
**for** $k = 1$ to $K$ **do**
    Update $\boldsymbol{\lambda}_k$ by optimizing problem (3).
    Compute the composite gradient $\boldsymbol{g}_k = \nabla \boldsymbol{f}(\boldsymbol{x}_k, \boldsymbol{\xi})^T \boldsymbol{\lambda}_k$.
    Update $\boldsymbol{x}_{k+1} = \boldsymbol{x}_k - \eta_k \boldsymbol{g}_k$.
**end for**

---

It is worth noting that in practice, the use of the weighted sum method with equal weights usually implies the a priori assumption that "all objectives are equally important", i.e., all objectives are expected to perform not bad. However, this example shows that even in a convex optimization problem, the equal-weight approach cannot guarantee an obtained Pareto front point with balanced performance. Next, we will develop an algorithm that aims to converge stably to a Pareto front point with balanced performance.

## 4 ROBUSTNESS-DRIVEN GRADIENT DESCENT

In this section, we fisrt transform an SMOO problem into a min-max problem and establish the theoretical correspondences between the two problems. Based on this, we then develop a robustness-driven gradient descent for SMOO and theoretically analyze the performance of the algorithm in both convex and non-convex settings.

### 4.1 ROBGRAD DEVELOPMENT

In order to obtain the Pareto front point with performance equilibrium for problem (1) from a global perspective, we try to minimize the worst-case objective value for problem (1) as follows.

$$\min_{\boldsymbol{x} \in \mathbb{R}^n} \max_{\boldsymbol{\lambda} \in \Lambda} \boldsymbol{\lambda}^T \boldsymbol{f}(\boldsymbol{x}) - \frac{\alpha}{2} \|\nabla \boldsymbol{f}(\boldsymbol{x})^T \boldsymbol{\lambda}\|^2. \tag{2}$$

where the multiplier $\alpha \geqslant 0$ and the second term $\|\nabla \boldsymbol{f}(\boldsymbol{x})^T \boldsymbol{\lambda}\|^2$ is used to ensure the non-dominance of the obtained objective vector in convex settings and the Pareto stationarity of the results obtained in non-convex settings (Fliege & Svaiter, 2000). We establish the connection between problem (2) and problem (1) as the following propositions, the proofs of which are provided in Section B.

**Proposition 1** (Convex). *In convex settings, if the objective functions $\boldsymbol{f}(\boldsymbol{x})$ in problem (2) are differentiable and $L$-smoothness, suppose that $\alpha < \frac{1}{L}$ and the optimal solution set $\mathcal{S}_{opt}$ of problem (2) is nonempty, then $\mathcal{S}_{opt} \subset Y_{WN}$ and there exists an efficient solution $\boldsymbol{x}^* \in Y_N$ of problem (1) such that $\boldsymbol{x}^* \in \mathcal{S}_{opt}$.*

**Proposition 2** (Non-convex). *In non-convex settings, if the objective functions $\boldsymbol{f}(\boldsymbol{x})$ in problem (2) are differentiable and $L$-smoothness, when $\alpha < \frac{1}{L}$, then an optimal solution $\boldsymbol{x}^*$ of problem (2) is Pareto stationary for problem (1).*

The above two theorems show the essential connection between the solution of problem (2) and the solution set of the SMOO problem (1) in terms of both convex and non-convex scenarios. Our approach, constructed by problem transformation, is deeply coupled to the SMOO problem and is able to deal with both convex and non-convex scenarios from a global perspective. (Weakly) Efficient solutions are obtained in convex optimization, and Pareto stationary solutions are guaranteed in non-convex optimization.

To solve problem (2), we introduce a primal-dual approach, which takes a dual ascent step to update the variables $\boldsymbol{\lambda}$ and a descent step to update the decision variables $\boldsymbol{x}$. Specifically, in stochastic environments, $\boldsymbol{\lambda}_k$ are obtained at $k$-th round by optimizing the following problem:

$$\min_{\boldsymbol{\lambda} \in \Lambda} -\boldsymbol{\lambda}^T \boldsymbol{f}(\boldsymbol{x}_k, \boldsymbol{\xi}) + \frac{\alpha_k}{2} \|\nabla \boldsymbol{f}(\boldsymbol{x}_k, \boldsymbol{\xi})^T \boldsymbol{\lambda}\|^2 + \beta_k \mathcal{R}_k, \tag{3}$$

where $\beta_k > 0$ is a positive stepsize and $\mathcal{R}_k = \|\boldsymbol{\lambda} - \boldsymbol{\lambda}_{k-1}\|^2$. Then, $\boldsymbol{x}$ is update by $\boldsymbol{x}_{k+1} = \boldsymbol{x}_k - \eta_k \nabla \boldsymbol{f}(\boldsymbol{x}_k, \boldsymbol{\xi})^T \boldsymbol{\lambda}_k$, where $\eta_k$ is a positive stepsize. The complete algorithm is summarized in Algorithm 1. In fact, the objective function of problem (3) can be viewed as consist-

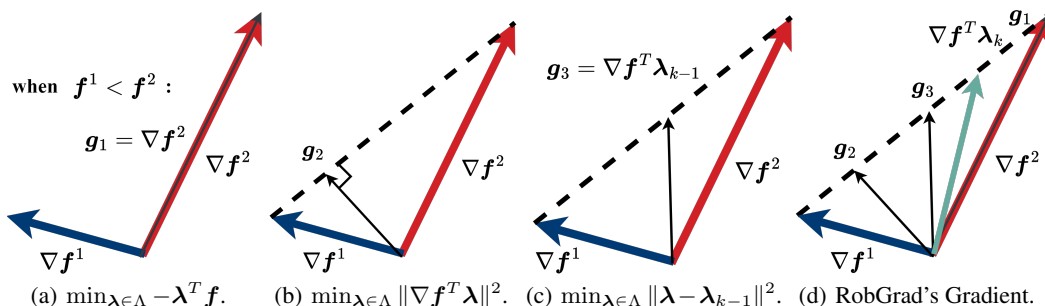

(a) $\min_{\boldsymbol{\lambda}\in\Lambda} -\boldsymbol{\lambda}^T\boldsymbol{f}$. (b) $\min_{\boldsymbol{\lambda}\in\Lambda} \|\nabla\boldsymbol{f}^T\boldsymbol{\lambda}\|^2$. (c) $\min_{\boldsymbol{\lambda}\in\Lambda} \|\boldsymbol{\lambda}-\boldsymbol{\lambda}_{k-1}\|^2$. (d) RobGrad's Gradient.

Figure 3: Geometric interpretation of RobGrad on an SMOO problem with two objectives, where $\boldsymbol{f} := \boldsymbol{f}(\boldsymbol{x}_k, \boldsymbol{\xi})$ and $\nabla\boldsymbol{f} := \nabla\boldsymbol{f}(\boldsymbol{x}_k, \boldsymbol{\xi})$. For the three terms of the objective in problem (3), Figures 3(a)-3(d) respectively display gradients ($\boldsymbol{g}_1, \boldsymbol{g}_2$ and $\boldsymbol{g}_3$) obtained by independently optimizing each individual term. Figure 3(d) shows that the gradient $\nabla\boldsymbol{f}^T\boldsymbol{\lambda}_k$ obtained from RobGrad. It can be seen that $\boldsymbol{g}_1$ prioritizes improving the objective with lagging performance, $\boldsymbol{g}_2$ focuses on eliminating gradient conflicts and $\boldsymbol{g}_3$ prevents sudden changes in weights, while RobGrad balances these roles.

ing of three terms: an objective balance term $-\boldsymbol{\lambda}^T\boldsymbol{f}(\boldsymbol{x}_k, \boldsymbol{\xi})$, a gradient conflict mitigation term $\|\nabla\boldsymbol{f}(\boldsymbol{x}_k, \boldsymbol{\xi})^T\boldsymbol{\lambda}\|^2$, and a weights smoothing term $\mathcal{R}_k$. Figure 3 provides a geometric interpretation of them and we explore their role in practice in Section 5.

## 4.2 CONVERGENCE ANALYSIS

We provide performance analyses of Algorithm 1 based on the following two basic assumptions, which are commonly used in complexity analysis for stochastic gradient algorithms (Mahdavi et al., 2013; Zhou et al., 2022; Fernando et al., 2023; Xu et al., 2025).

**Assumption 1.** *For the iterative sequence $\{\boldsymbol{x}_k\}$, the variance of each objective function is bounded by $\mathbb{E}_{\boldsymbol{\xi}}[\|f^i(\boldsymbol{x}_k, \boldsymbol{\xi}) - f^i(\boldsymbol{x}_k)\|^2] \leqslant \sigma_1^2, \forall i \in [p]$.*

**Assumption 2.** *The objective functions have subgradients and they are L-smoothness and G-Lipschitz i.e., $\|\nabla f^i(\boldsymbol{x})\| \leqslant G, \forall i \in [p]$. For the iterative sequence $\{\boldsymbol{x}_k\}$, each objective function variance is bounded by $\mathbb{E}_{\boldsymbol{\xi}}[\|\nabla f^i(\boldsymbol{x}_k, \boldsymbol{\xi}) - \nabla f^i(\boldsymbol{x}_k)\|^2] \leqslant \sigma_2^2, \forall i \in [p]$.*

Based on the above assumptions, we establish the non-asymptotic convergence bounds for Algorithm 1 as the following theorems, the proofs of which are provided in Section C.

**Theorem 1** (Convex). *Under Assumptions 1 and 2, when the objective functions are convex with a bound as $|f^i(\boldsymbol{x})| \leqslant F$, $i \in [p]$, the distance from sequence $\{\boldsymbol{x}_k\}$ generated by Algorithm 1 with $\beta_k \geqslant 0$ to an optimal solution $\boldsymbol{x}^*$ in (2) is bounded, i.e., $\|\boldsymbol{x}_k - \boldsymbol{x}^*\| \leqslant D$, then with a probability $(1-\delta)$ we have*

$$\frac{1}{K}\sum_{k=1}^{K}\mathbb{E}_{\boldsymbol{\xi}}[\boldsymbol{\lambda}^{*T}(\boldsymbol{f}(\boldsymbol{x}_k) - \boldsymbol{f}(\boldsymbol{x}^*))] \leqslant \frac{D^2}{2K\eta_k} + \frac{\sqrt{2}}{K}\sum_{k=1}^{K}\beta_k + \frac{\sigma_2^2 + G^2}{K}\sum_{k=1}^{K}(\eta_k + \alpha_k)$$

$$+ \frac{2\sqrt{2(D^2G^2 + F^2)\ln\frac{1}{\delta}}}{\sqrt{K}}.$$

*Setting $\eta_k = \mathcal{O}(1/\sqrt{k})$, $\alpha_k \leqslant \mathcal{O}(1/\sqrt{k})$ and $\beta_k \leqslant \mathcal{O}(1/\sqrt{k})$, the convergence rate is $\mathcal{O}(1/\sqrt{K})$.*

**Theorem 2** (Non-convex). *Under Assumptions 1 and 2, suppose the objective functions are bounded by $|f^i(\boldsymbol{x})| \leqslant F$, $i \in [p]$, for the Algorithm 1 with $\beta_k > 0$, we have*

$$\frac{1}{K}\sum_{k=1}^{K}\mathbb{E}_{\boldsymbol{\xi}}[\min_{\boldsymbol{\lambda}_k^*\in\Lambda}\|\nabla\boldsymbol{f}(\boldsymbol{x}_k)^T\boldsymbol{\lambda}_k^*\|^2] \leqslant \frac{F\sqrt{2C_1}}{K\eta_K}\sum_{k=2}^{K}\frac{1}{\sqrt{\beta_k}} + \frac{F\sqrt{C_2}}{K\eta_K}\sum_{k=2}^{K}\frac{\sqrt{\alpha_k}}{\sqrt{\beta_k}} + \frac{LC_2}{K}\sum_{k=1}^{K}\eta_k$$

$$+ \frac{2pG\sigma_2\sqrt{2C_1}}{K}\sum_{k=1}^{K}\frac{1}{\sqrt{\beta_k}} + \frac{2pG\sigma_2\sqrt{C_2}}{K}\sum_{k=1}^{K}\frac{\sqrt{\alpha_k}}{\sqrt{\beta_k}} + \frac{2F}{K\eta_K}.$$

*where $C_1 = \sigma_1 + F$ and $C_2 = \sigma_2^2 + G^2$. When $\eta_k = \mathcal{O}(1/\sqrt{k})$, $\alpha_k = \mathcal{O}(1/k^{1/3})$ and $\beta_k = \mathcal{O}(k^{4/3})$, then the convergence rate is $\mathcal{O}(1/K^{1/6})$.*

Thus, from the above Theorems and Figure 3, for non-convex settings, by adjusting $\alpha_k$ and $\beta_k$ in (3), RobGrad tries to improve the lagging performance objective in the early stage while avoiding gradient conflicts, and stabilizes the convergence in the later stage.

## 5 EXPERIMENT

In this section, we present extensive empirical results of RobGrad, including a bi-objective synthesis example and multi-task learning. We conduct experiments to answer the following questions:
**Q1:** Does our algorithm already address the gap mentioned in Section 1?
**Q2:** Does our algorithm provide better model performance in multi-task learning?
**Q3:** How does each component of problem (3)'s objective contribute to the performance in practice?

**Baseline.** We use the following 15 baselines. (1) **Single-task**: optimizing individual objectives independently; (2) **Uniform**: optimizing a uniformly weighted sum of individual objectives; (3) **Uncertainty Weighting (UW)** (Kendall et al., 2018): assigning weights by moscedastic uncertainty for each objective; (4) **Dymanic Weight Average (DWA)** (Liu et al., 2019): adjusting the weights according to the rate of change of the multi-objective loss values; (5) **MGDA** (Sener & Koltun, 2018): obtaining weights by minimizing the norm of convex combinations of multiple gradients; (6) **GradDrop** (Chen et al., 2020): forcing the sign consistency among multiple gradients; (7) **PCGrad** (Yu et al., 2020): obtaining non-conflicting descent directions by a gradient projection strategy; (8) **CAGrad** (Liu et al., 2021a): finding the best optimization direction in a sphere around an average gradient that maximizes the worse local improvement among multiple objective; (9) **IMTL-G** (Liu et al., 2021b): choosing an optimization direction such that it projects equally on each objective gradient direction and on the average gradient direction; (10) **Nash-MTL** (Navon et al., 2022): determining an optimization direction by a bargaining game; (11) **Random Loss Weighting (RLW)** (Lin et al., 2022): sampling gradient weights from a normal distribution; (12) **FAMO** (Liu et al., 2023): updating weights by losses history; (13) **FairGrad** (Maheshwari & Perrot, 2023): learning group-specific weights iteratively through a reweighting scheme; (14) **SDMGrad** (Xiao et al., 2023): obtaining an optimization direction by maximizing the minimum decrease of all objectives; (15) **PSMGD** (Xu et al., 2025): updating weights periodically to speed up the optimization process.

**Datasets.** We evaluated our algorithm on a synthetic example and 5 multi-task datasets. The synthetic example (Liu et al., 2021a) is a non-convex bi-objective optimization problem. The multi-task datasets are CityScapes (Cordts et al., 2016), CelebA (Liu et al., 2015), NYU-v2 (Silberman et al., 2012), Multi-MNIST (Sener & Koltun, 2018) and QM-9 (Blum & Reymond, 2009). Due to the lack of space, more experimental results and details are provided in Section D.

### 5.1 SYNTHETIC EXAMPLE

To answer question **Q1**, we use a synthetic bi-objective optimization problem (Liu et al., 2021a) to evaluate the performance of RobGrad, where this problem is presented in detail in the supplementary material. Figure 1 plots the optimization trajectories of different algorithms based on five different initial points. For poor initial points, Uniform and FairGrad in Figures 1(a) and 1(b) fall into the local optimal point for a long time. SDMGrad and PSMGD in Figures 1(c) and 1(d) have the typical characteristics of gradient-based methods, where the optimization direction is decided based only on the instantaneous gradient, and the final optimization results rely on initial points. The results for more baseline algorithms are similar and are provided in the supplementary material. Compared to the above algorithms, RobGrad is designed based on a global perspective and is robust to optimization results for different initial points. Specifically, in Figure 1(e), RobGrad not only escapes the local optimum quickly but also always obtains the Pareto front point with balanced performance. This experimental result, Theorem 1 and 2 positively answer **Q1**.

### 5.2 MULTI-TASK LEARNING

To answer question **Q2**, we evaluated the performance of the algorithms on the CityScapes, CelebA and NYU-v2. CityScapes is an urban street scene dataset that involves 2 tasks: 7-class semantic

Table 1: The results on CityScapes (2 tasks) and CelebA (40 tasks) datasets. The mean of each metric (↑ indicates higher better and ↓ indicates lower better) is reported over 3 independent runs. The best average result is marked in bold. **MR** and $\Delta M\%$ are the main metrics.

| Method | CityScapes | | | | | | CelebA | |
| | Segmentation | | Depth | | MR ↓ | $\Delta M\%$ ↓ | MR ↓ | $\Delta M\%$ ↓ |
| | mIoU ↑ | Pix Acc ↑ | Abs Err ↓ | Rel Err ↓ | | | | |
| Single-task | 74.01 | 93.16 | 0.0125 | 27.77 | | | | |
| Uniform | 75.18 | 93.49 | 0.0155 | 46.77 | 9.50 | 22.60 | 8.40 | 6.46 |
| UW | 72.02 | 92.85 | 0.0140 | **30.13** | 9.00 | 5.89 | 6.83 | 5.68 |
| DWA | 75.24 | 93.52 | 0.0160 | 44.37 | 9.25 | 21.45 | 6.88 | 6.27 |
| MGDA | 68.84 | 91.54 | 0.0309 | 33.50 | 12.25 | 44.14 | 10.35 | 10.15 |
| GradDrop | 75.27 | 93.53 | 0.0157 | 47.54 | 8.75 | 23.73 | 8.30 | 6.99 |
| PCGrad | 75.13 | 93.48 | 0.0154 | 42.07 | 9.75 | 18.29 | 6.50 | 6.87 |
| CAGrad | 75.16 | 93.48 | 0.0141 | 37.60 | 8.50 | 11.64 | 7.90 | 6.09 |
| IMTL-G | 75.33 | 93.49 | 0.0135 | 38.41 | 6.25 | 11.10 | 6.25 | 4.61 |
| Nash-MTL | 75.41 | 93.66 | **0.0129** | 35.02 | 3.50 | 6.82 | 8.70 | 4.52 |
| RLW | 74.57 | 93.41 | 0.0158 | 47.79 | 12.50 | 24.38 | 8.00 | 5.55 |
| FAMO | 74.54 | 93.29 | 0.0145 | 32.59 | 9.25 | 8.13 | **6.03** | 4.26 |
| FairGrad | 75.72 | 93.68 | 0.0134 | 32.25 | **2.00** | **5.18** | 6.65 | 3.25 |
| SDMGrad | 74.53 | 93.52 | 0.0137 | 34.01 | 7.50 | 7.79 | × | × |
| PSMGD | 74.90 | 93.37 | 0.0135 | 35.78 | 8.50 | 8.76 | 6.65 | 5.11 |
| RobGrad | **75.91** | **93.72** | 0.0134 | 32.81 | **2.00** | 5.55 | 7.58 | **3.09** |

segmentation and depth estimation. CelebA is a large-scale face attributes dataset with over 200K images of celebrities, each with 40 different attributes. NYU-v2 is an indoor scene dataset that contains 3 tasks: 13-class semantic segmentation, depth estimation, and surface normal estimation.

We follow the same experimental setting of Xiao et al. (2023) and Xu et al. (2025). Besides task specific metrics, we use two comprehensive metrics to evaluate the overall performance of algorithms on multiple tasks. **(1) Mean Rank (MR)**: the average rank of each algorithm in the metrics. **(2)** The average per-task performance drop of an algorithm $a$ versus the single-task baseline $b$: $\Delta M\% = \frac{1}{N_m} \sum_{n=1}^{N_m} (-1)^{l_n} (m_{a,n} - m_{b,n})/m_{b,n} \times 100$, where $N_m$ is the number of metrics, $m_{a,n}$ and $m_{b,n}$ are respectively the $n$-th metric values for algorithms $a$ and single-task $b$, $l_n = 1$ if higher $n$-th metric is better and 0 otherwise.

The parameters $\alpha_k$ and $\beta_k$ are selected via grid search on $\{0.0001, 0.001, \cdots, 1.0\}$ for RobGrad. We report the results in Tables 1 and 2. The results show that RobGrad has competitive or improved performance compared to previous methods. In particular, new SOTA results are achieved on the comprehensive metrics **MR** and $\Delta M\%$ on the CityScapes, CelebA and NYU-v2 datasets. Specifically, according to metrics MR and $\Delta M\%$, when RobGrad achieves the lowest average ranking and the most comprehensive performance improvement for the single-task algorithm, this means that RobGrad can balance multiple objectives better than other baselines such that the results for each objective are not bad. This positively answers **Q2**.

## 5.3 ROLE OF COMPONENTS

To answer **Q3**, we explore the roles of the three components of the objective function in problem (3) on the Multi-MNIST dataset, which contains 2 tasks: simultaneously classifying the digit on the top-left (task-L) and the bottom-right (task-R) separately. We follow the setup of Sener & Koltun (2018) and use the LeNet network architecture.

We denote the first term $-\boldsymbol{\lambda}^T \boldsymbol{f}(\boldsymbol{x}_k, \boldsymbol{\xi})$ as $\mathcal{C}_1$, the second term $\|\nabla \boldsymbol{f}(\boldsymbol{x}_k, \boldsymbol{\xi})^T \boldsymbol{\lambda}\|^2$ as $\mathcal{C}_2$ and third term $\mathcal{R}_k$ as $\mathcal{C}_3$. We conducted experiments for RobGrad, designing seven groups of controlled trials: three groups retained only a single component, three groups removed one component individually, and one group with all components intact.

Table 2: The results on the NYU-v2 dataset (3 tasks). The mean of each metric (↑ indicates higher better and ↓ indicates lower better) is reported over 3 independent runs. The best average result is marked in bold. **MR** and $\mathbf{\Delta M}\%$ are the main metrics.

| Method | Segmentation | | Depth | | Surface Normal | | | | | MR ↓ | $\Delta M\%$ ↓ |
| | | | | | Angle Distance ↓ | | Within $t°$ ↑ | | | | |
| | mIoU ↑ | Pix Acc ↑ | Abs Err ↓ | Rel Err ↓ | Mean | Median | 11.25 | 22.5 | 30 | | |
|---|---|---|---|---|---|---|---|---|---|---|---|
| Single-task | 38.30 | 63.76 | 0.6754 | 0.2780 | 25.01 | 19.21 | 30.14 | 57.20 | 69.15 | | |
| Uniform | 39.29 | 65.33 | 0.5493 | 0.2263 | 28.15 | 23.96 | 22.09 | 47.50 | 61.08 | 11.44 | 5.59 |
| UW | 36.87 | 63.17 | 0.5446 | 0.2260 | 27.04 | 22.61 | 23.54 | 49.05 | 63.65 | 10.67 | 4.05 |
| DWA | 39.11 | 65.31 | 0.5510 | 0.2285 | 27.61 | 23.18 | 24.17 | 50.18 | 62.39 | 10.78 | 3.57 |
| MGDA | 30.47 | 59.90 | 0.6070 | 0.2555 | 24.88 | 19.45 | 29.18 | 56.88 | 69.36 | 8.67 | 1.38 |
| GradDrop | 39.39 | 65.12 | 0.5455 | 0.2279 | 27.48 | 22.96 | 23.38 | 49.44 | 62.87 | 10.11 | 3.58 |
| PCGrad | 38.06 | 64.64 | 0.5550 | 0.2325 | 27.41 | 22.80 | 23.86 | 49.83 | 63.14 | 11.33 | 3.97 |
| CAGrad | 39.79 | 65.49 | 0.5486 | 0.2250 | 26.31 | 21.58 | 25.61 | 52.36 | 65.58 | 7.56 | 0.20 |
| IMTL-G | 39.35 | 65.60 | 0.5426 | 0.2256 | 26.02 | 21.19 | 26.20 | 53.13 | 66.24 | 6.89 | -0.76 |
| Nash-MTL | 40.13 | 65.93 | 0.5261 | 0.2171 | 25.26 | 20.08 | 28.40 | 55.47 | 68.15 | 4.78 | -4.04 |
| RLW | 37.17 | 63.77 | 0.5759 | 0.2410 | 28.27 | 24.18 | 22.26 | 47.05 | 60.62 | 14.11 | 7.78 |
| FAMO | 38.88 | 64.90 | 0.5474 | 0.2194 | 25.06 | 19.57 | 29.21 | 56.61 | 68.98 | 5.78 | -4.10 |
| FairGrad | 39.74 | **66.01** | 0.5377 | 0.2236 | 24.84 | 19.60 | 29.26 | 56.58 | 69.16 | 3.67 | -4.66 |
| SDMGrad | **40.47** | 65.90 | **0.5225** | **0.2084** | 25.07 | 19.99 | 28.54 | 55.74 | 68.53 | 4.00 | -4.84 |
| PSMGD | 35.44 | 63.78 | 0.5494 | 0.2369 | 24.83 | **18.89** | 30.68 | 58.00 | 69.84 | 6.56 | -3.62 |
| RobGrad | 38.77 | 65.01 | 0.5422 | 0.2235 | **24.79** | **18.89** | **30.71** | **58.01** | **70.03** | **3.56** | **-5.53** |

We report the experimental results in Table 3 that includes the following three metrics: (1) **Balancing Angle (BA)**: the angle between the accuracy vector of the two tasks and the benchmark vector $e$; (2) **Convergence Time (CT)**: the time step when the average accuracy of the two tasks exceeds $90\%$ for the first time; (3) **Peak Accuracy (PA)**: the highest average accuracy of the two tasks.

Table 3: The results of experiments on the Multi-MNIST dataset. The mean of each metric (↑ indicates higher better and ↓ indicates lower better) is reported over 3 independent runs. The best average result is marked in bold.

| Method | Components | | | BA ↓ | CT ↓ | PA ↑ |
| | $\mathcal{C}_1$ | $\mathcal{C}_2$ | $\mathcal{C}_3$ | $(a°)$ | (min) | |
|---|---|---|---|---|---|---|
| RobGrad only $\mathcal{C}_1$ | ✓ | | | 0.0543 | 30.50 | 0.9493 |
| RobGrad only $\mathcal{C}_2$ | | ✓ | | 0.2785 | 19.09 | 0.9567 |
| RobGrad only $\mathcal{C}_3$ | | | ✓ | 0.3473 | 23.87 | 0.9569 |
| RobGrad w/o $\mathcal{C}_1$ | | ✓ | ✓ | 0.4744 | 15.82 | 0.9601 |
| RobGrad w/o $\mathcal{C}_2$ | ✓ | | ✓ | 0.0745 | 16.79 | 0.9613 |
| RobGrad w/o $\mathcal{C}_3$ | ✓ | ✓ | | 0.0358 | 15.80 | 0.9604 |
| RobGrad | ✓ | ✓ | ✓ | **0.0297** | **14.29** | **0.9631** |

The results in Table 3 show that the BAs of the algorithms with $\mathcal{C}_1$ are low, reflecting the balancing effect of $\mathcal{C}_1$ on the objectives. The CTs of the algorithms with $\mathcal{C}_2$ are low, revealing the ability of $\mathcal{C}_2$ to mitigate gradient conflicts. RobGrad combines $\mathcal{C}_1$, $\mathcal{C}_2$ and $\mathcal{C}_3$ to further improve the model performance and achieve the highest average accuracy.

# 6 CONLUSION

In this paper, we transform the stochastic multi-objective optimization (SMOO) problem into a min-max problem to obtain the Pareto front point with performance balance, and establish a theoretical correspondence between this min-max problem and the SMOO problem. On this basis, a Robustness-Driven Gradient Descent (RobGrad) algorithm is proposed and its non-asymptotic convergence bounds are proved. Extensive experiments show that RobGrad significantly improves the comprehensive performance of the multi-task learning models compared to existing algorithms.

ETHICS STATEMENT

The authors declare that they have no known competing financial interests or personal relationships that could have appeared to influence the work reported in this paper.

REPRODUCIBILITY STATEMENT

The implementation of the algorithm proposed in this paper and all experiments are reproducible. We have uploaded the anonymized source code to the anonymized link (https://anonymous.4open.science/r/RobGrad). The assumptions and proofs of the propositions and theorems are detailed in Section 4, B and C. The dataset sources and experimental settings are fully described in Section 5 and D.

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

# RobGrad: Robustness-Driven Gradient Descent for Stochastic Multi-Objective Optimization (Supplementary material)

This supplementary material is organized as follows:

- **Appendix A** introduce related works on stochastic minimax optimization.

- **Appendix B** proves the theoretical correspondences between problem (2) and (1).

- **Appendix C** proves the non-asymptotic convergence bounds of RobGrad.

- **Appendix D** provides more experimental details and results.

- **Appendix E** discloses the use of Large Language Models (LLMs).

## A  RELATED WORKS ON STOCHASTIC MINIMAX OPTIMIZATION

Our RobGrad algorithm is designed based on the minimax problem (2), so stochastic minimax optimization is also related to our work. Many algorithms have been designed to address the different properties of objective functions, such as convex-concave (Nemirovski et al., 2009), strongly-convex-strongly-concave (Palaniappan & Bach, 2016; Zhao, 2022; Laguel et al., 2023), nonconvex-concave (Rafique et al., 2018; Lin et al., 2020; Boţ & Böhm, 2023), nonconvex-nonconcave (Yang et al., 2020) and Nonconvex-Polyak-Łojasiewicz (Yang et al., 2022; Laguel et al., 2024). However, due to the following reasons in two aspects, none of the above methods are applicable to solving problem (2).

For the first aspect, due to the second term $-\frac{\alpha}{2}\|\nabla \boldsymbol{f}(\boldsymbol{x})^T \boldsymbol{\lambda}\|^2$ in problem (2), problem (2) is a nonconvex-concave problem. However, in the above methods tailored for non-convex minimax problems, method design often relies on the assumption that the gradient of the objective function is bounded or Lipschitz-smooth. This implicitly requires imposing a bound on the third-order derivatives of the objectives in the SMOO problem (1), which is often overly stringent in practice. For the second aspect, the convergence analysis of the above methods is primarily concerned with the primal/dual gap and the notion of a nearly stationary point. However, the convergence analysis in this work is conducted for the Pareto optimality in the SMOO problem (1). Therefore, based on problem (2), we proposes the RobGrad algorithm to address the SMOO problem (1).

## B  MISSING PROOFS FOR THE PARETO OPTIMALITY OF THE MIN-MAX PROBLEM'S SOLUTION

Before proving Propositions 1 and 2, we introduce the following lemma.

**Lemma 1.** *Under the same assumptions as in Proposition 1, we have $\|\nabla \boldsymbol{f}(\boldsymbol{x}^*)^T \boldsymbol{\lambda}^*\| = 0$.*

*Proof.* Let $(\boldsymbol{x}^*, \boldsymbol{\lambda}^*)$ be an optimal solution of problem 2, then $\boldsymbol{x}^*$ satisfies the first-order condition as follows.

$$\nabla \boldsymbol{f}(\boldsymbol{x}^*)^T \boldsymbol{\lambda}^* - \frac{\alpha}{2} \nabla_{\boldsymbol{x}}(\|\nabla \boldsymbol{f}(\boldsymbol{x}^*)^T \boldsymbol{\lambda}^*\|^2) = \nabla \boldsymbol{f}(\boldsymbol{x}^*)^T \boldsymbol{\lambda}^* - \alpha \nabla_{\boldsymbol{x}}(\nabla \boldsymbol{f}(\boldsymbol{x}^*)^T \boldsymbol{\lambda}^*)^T \nabla \boldsymbol{f}(\boldsymbol{x}^*)^T \boldsymbol{\lambda}^*$$

$$= \nabla \boldsymbol{f}(\boldsymbol{x}^*)^T \boldsymbol{\lambda}^* - \alpha(\sum_{i=1}^{p} \boldsymbol{\lambda}^i \nabla^2 \boldsymbol{f}(\boldsymbol{x}^*)^T \boldsymbol{\lambda}^*)^T \nabla \boldsymbol{f}(\boldsymbol{x}^*)^T \boldsymbol{\lambda}^*$$

$$= \boldsymbol{0}.$$

We then prove Lemma 1 by contradiction. Suppose $\nabla \boldsymbol{f}(\boldsymbol{x}^*)^T \boldsymbol{\lambda}^* \neq \boldsymbol{0}$ and let $\boldsymbol{d} := \nabla \boldsymbol{f}(\boldsymbol{x}^*)^T \boldsymbol{\lambda}^*$, we have

$$\boldsymbol{d}^T \boldsymbol{d} - \alpha \boldsymbol{d}^T (\sum_{i=1}^{p} \boldsymbol{\lambda}^i \nabla^2 \boldsymbol{f}(\boldsymbol{x}^*)^T \boldsymbol{d} = \|\boldsymbol{d}\|^2 - \alpha \boldsymbol{d}^T A^T \boldsymbol{d} = \boldsymbol{d}^T (I - \alpha H)^T \boldsymbol{d} = 0, \quad (4)$$

where $H = \sum_{i=1}^{p} \boldsymbol{\lambda}^i \nabla^2 \boldsymbol{f}(\boldsymbol{x}^*)^T \boldsymbol{\lambda}^*$.

According to $L$-smoothness, the eigenvalues $\varepsilon$ of the Hessian matrix $H$ are upper bounded by $L$, i.e., $\max \varepsilon \leqslant L$. Since $\alpha < \frac{1}{L}$, then we have $\alpha H < I$, i.e., $\alpha H \neq I$. Therefore, $\boldsymbol{d} = \boldsymbol{0}$ due to (4). $\qquad \square$

## B.1 PROOF OF PROPOSITION 1

*Proof.* According to Lemma 1, we have $\nabla \boldsymbol{f}(\boldsymbol{x}^*)^T \boldsymbol{\lambda}^* = \boldsymbol{0}$. Thus, $\boldsymbol{x}^*$ is a weakly efficient solution of the SMOO problem by convexity, which means that there does not exist an $\boldsymbol{x} \in \mathbb{R}^n$ such that $\boldsymbol{f}(\boldsymbol{x}) < \boldsymbol{f}(\boldsymbol{x}^*)$.

Then there exists an efficient solution $\boldsymbol{x}' \in \mathbb{R}^n$ such that $\boldsymbol{f}(\boldsymbol{x}') \leqslant \boldsymbol{f}(\boldsymbol{x}^*)$ and there exists $i \in [p]$ such that $\boldsymbol{f}^i(\boldsymbol{x}') < \boldsymbol{f}^i(\boldsymbol{x}^*)$, which implies that

$$\forall \boldsymbol{\lambda} \in \Lambda, \boldsymbol{\lambda}^T \boldsymbol{f}(\boldsymbol{x}') \leqslant \boldsymbol{\lambda}^T \boldsymbol{f}(\boldsymbol{x}^*). \quad (5)$$

Since $\boldsymbol{x}^*$ is a weakly efficient solution, we set $I := \{i \in [p] \mid \boldsymbol{f}^i(\boldsymbol{x}') = \boldsymbol{f}^i(\boldsymbol{x}^*)\}$ and $J := \{j \in [p] \mid \boldsymbol{f}^j(\boldsymbol{x}') < \boldsymbol{f}^j(\boldsymbol{x}^*)\}$.

Then, there exists a weight $\boldsymbol{\lambda}'$ such that the optimization direction $\nabla \boldsymbol{f}(\boldsymbol{x}^*)^T \boldsymbol{\lambda}'$ improves the objective function value of the SMOO problem, i.e., $\boldsymbol{x}' = \boldsymbol{x}^* - \delta \nabla \boldsymbol{f}(\boldsymbol{x}^*)^T \boldsymbol{\lambda}'$, where for all $j \in J$, $\boldsymbol{\lambda}'^j > 0$ and $\delta > 0$ is arbitrarily small. Here, we have

$$\boldsymbol{\lambda}'^T \boldsymbol{f}(\boldsymbol{x}') < \boldsymbol{\lambda}'^T \boldsymbol{f}(\boldsymbol{x}^*). \quad (6)$$

Since $(\boldsymbol{x}^*, \boldsymbol{\lambda}^*)$ is the optimal solution of problem 2, then

$$\boldsymbol{\lambda}^{*T} \boldsymbol{f}(\boldsymbol{x}^*) - \frac{\alpha}{2} \|\nabla \boldsymbol{f}(\boldsymbol{x}^*)^T \boldsymbol{\lambda}^*\|^2 \leqslant \boldsymbol{\lambda}^{*T} \boldsymbol{f}(\boldsymbol{x}') - \frac{\alpha}{2} \|\nabla \boldsymbol{f}(\boldsymbol{x}')^T \boldsymbol{\lambda}^*\|^2 \leqslant \boldsymbol{\lambda}^{*T} \boldsymbol{f}(\boldsymbol{x}'). \quad (7)$$

Thus,

$$\boldsymbol{\lambda}^{*T} \boldsymbol{f}(\boldsymbol{x}^*) \leqslant \boldsymbol{\lambda}^{*T} \boldsymbol{f}(\boldsymbol{x}'). \quad (8)$$

Combining (5) and (8), we have $\boldsymbol{\lambda}^{*T} \boldsymbol{f}(\boldsymbol{x}^*) = \boldsymbol{\lambda}^{*T} \boldsymbol{f}(\boldsymbol{x}')$, which implies that for all $j \in J$, $\boldsymbol{\lambda}^{*j} = 0$. Therefore, for (7), we have $-\frac{\alpha}{2} \|\nabla \boldsymbol{f}(\boldsymbol{x}^*)^T \boldsymbol{\lambda}^*\|^2 \leqslant -\frac{\alpha}{2} \|\nabla \boldsymbol{f}(\boldsymbol{x}')^T \boldsymbol{\lambda}^*\|^2$. Since $\alpha > 0$, we have $\|\nabla \boldsymbol{f}(\boldsymbol{x}')^T \boldsymbol{\lambda}^*\|^2 \leqslant 0$, i.e., $\|\nabla \boldsymbol{f}(\boldsymbol{x}')^T \boldsymbol{\lambda}^*\|^2 = 0$. Here, we have $F(\boldsymbol{x}^*, \boldsymbol{\lambda}^*) = F(\boldsymbol{x}', \boldsymbol{\lambda}^*)$, which means that $\boldsymbol{x}'$ is an optimal solution of the min-max problem. Thus, if the set of optimal solutions to the min-max problem is non-empty, then it contain an efficient solution of the SMOO problem.

$\qquad \square$

## B.2 PROOF OF PROPOSITION 2

*Proof.* Let $(\boldsymbol{x}^*, \boldsymbol{\lambda}^*)$ be an optimal solution of problem 2, according to (1), we then have $\|\nabla \boldsymbol{f}(\boldsymbol{x}^*)^T \boldsymbol{\lambda}^*\| = 0$. Therefore, Proposition 2 holds. $\qquad \square$

# C   MISSING PROOFS FOR ROBGRAD'S NON-ASYMPTOTIC CONVERGENCE PROPERTY

## C.1   PROOF OF THEOREM 1

Before proving Theorem 1, we introduce the following lemmas.

**Lemma 2.** *Under the same assumptions as in Theorem 1, for a $\boldsymbol{\lambda} \in \Lambda$, we have the following inequality*

$$\mathbb{E}_{\boldsymbol{\xi}}[\|\boldsymbol{\lambda}_k^T \nabla \boldsymbol{f}(\boldsymbol{x}_k, \boldsymbol{\xi})\|^2] \leqslant 2\sigma_2^2 + 2G^2$$

*Proof.*

$$
\begin{aligned}
\mathbb{E}_{\boldsymbol{\xi}}[\|\boldsymbol{\lambda}^T \nabla \boldsymbol{f}(\boldsymbol{x}_k, \boldsymbol{\xi})\|^2] =& \mathbb{E}_{\boldsymbol{\xi}}[\|\boldsymbol{\lambda}^T \nabla \boldsymbol{f}(\boldsymbol{x}_k, \boldsymbol{\xi}) - \boldsymbol{\lambda}^T \nabla \boldsymbol{f}(\boldsymbol{x}_k) + \boldsymbol{\lambda}^T \nabla \boldsymbol{f}(\boldsymbol{x}_k)\|^2] \\
\leqslant& \mathbb{E}_{\boldsymbol{\xi}}[2\|\boldsymbol{\lambda}^T \nabla \boldsymbol{f}(\boldsymbol{x}_k, \boldsymbol{\xi}) - \boldsymbol{\lambda}^T \nabla \boldsymbol{f}(\boldsymbol{x}_k)\|^2 + 2\|\boldsymbol{\lambda}^T \nabla \boldsymbol{f}(\boldsymbol{x}_k)\|^2] \\
=& 2\mathbb{E}_{\boldsymbol{\xi}}[\sum_{i=1}^{p} \|\boldsymbol{\lambda}^i \nabla \boldsymbol{f}^i(\boldsymbol{x}_k, \boldsymbol{\xi}) - \boldsymbol{\lambda}^i \nabla \boldsymbol{f}^i(\boldsymbol{x}_k)\|^2] + 2\|\boldsymbol{\lambda}^T \nabla \boldsymbol{f}(\boldsymbol{x}_k)\|^2 \\
\leqslant& 2\sigma_2^2 + 2G^2
\end{aligned}
$$

$\square$

**Lemma 3.** *Under the same assumptions as in Theorem 1, we have the following inequality*

$$\mathbb{E}_{\boldsymbol{\xi}}\Big[\frac{1}{K}\sum_{k=1}^{K}(\boldsymbol{\lambda}_k^T \nabla \boldsymbol{f}(\boldsymbol{x}_k, \boldsymbol{\xi}))^T(\boldsymbol{x}_k - \boldsymbol{x}^*)\Big] \leqslant \frac{D^2}{2K\eta_k} + \frac{\sigma_2^2 + G^2}{K}\sum_{k=1}^{K}\eta_k$$

*Proof.* Using the update rule for $\boldsymbol{x}_k$, we have

$$\|\boldsymbol{x}_{k+1} - \boldsymbol{x}^*\|^2 \leqslant \|\boldsymbol{x}_k - \boldsymbol{x}^*\|^2 - 2\eta_k(\nabla \boldsymbol{f}(\boldsymbol{x}_k, \boldsymbol{\xi})^T \boldsymbol{\lambda}_k)^T(\boldsymbol{x}_k - \boldsymbol{x}^*) + \eta_k^2 \|\nabla \boldsymbol{f}(\boldsymbol{x}_k, \boldsymbol{\xi})^T \boldsymbol{\lambda}_k\|^2$$

Rearranging the above terms, it follows that

$$(\nabla \boldsymbol{f}(\boldsymbol{x}_k, \boldsymbol{\xi})^T \boldsymbol{\lambda}_k)^T(\boldsymbol{x}_k - \boldsymbol{x}^*) \leqslant \frac{\|\boldsymbol{x}_k - \boldsymbol{x}^*\|^2 - \|\boldsymbol{x}_{k+1} - \boldsymbol{x}^*\|^2}{2\eta_k} + \frac{\eta_k}{2}\|\nabla \boldsymbol{f}(\boldsymbol{x}_k, \boldsymbol{\xi})^T \boldsymbol{\lambda}_k\|^2$$

Summing the first term on the right-hand side of the above inequality from $k = 1$ to $K$ and setting $\eta_k$ ($\frac{1}{\eta_0^p} := 0$) be monotonically decreasing as $k$ increases:

$$
\begin{aligned}
\sum_{k=1}^{K} \frac{\|\boldsymbol{x}_k - \boldsymbol{x}^*\|^2 - \|\boldsymbol{x}_{k+1} - \boldsymbol{x}^*\|^2}{2\eta_k} \leqslant& \frac{1}{2}\sum_{k=1}^{K}\|\boldsymbol{x}_k - \boldsymbol{x}^*\|^2\Big(\frac{1}{\eta_k} - \frac{1}{\eta_{k-1}}\Big) \\
\leqslant& \frac{D^2}{2}\sum_{k=1}^{K}\Big(\frac{1}{\eta_k} - \frac{1}{\eta_{k-1}}\Big) \\
\leqslant& \frac{D^2}{2\eta_k}
\end{aligned}
$$

Thus,

$$\mathbb{E}_{\boldsymbol{\xi}}[\frac{1}{K}\sum_{k=1}^{K}(\nabla \boldsymbol{f}(\boldsymbol{x}_k, \boldsymbol{\xi})^T \boldsymbol{\lambda}_k)^T(\boldsymbol{x}_k - \boldsymbol{x}^*)]$$

$$\leqslant \mathbb{E}_{\boldsymbol{\xi}}[\frac{1}{K}\sum_{k=1}^{K}\frac{\|\boldsymbol{x}_k - \boldsymbol{x}^*\|^2 - \|\boldsymbol{x}_{k+1} - \boldsymbol{x}^*\|^2}{2\eta_k} + \frac{\eta_k}{2}\|\nabla \boldsymbol{f}(\boldsymbol{x}_k, \boldsymbol{\xi})^T \boldsymbol{\lambda}_k\|^2]$$

$$\leqslant \mathbb{E}_{\boldsymbol{\xi}}[\frac{D^2}{2K\eta_k} + \frac{1}{K}\sum_{k=1}^{K}\frac{\eta_k}{2}\|\nabla \boldsymbol{f}(\boldsymbol{x}_k, \boldsymbol{\xi})^T \boldsymbol{\lambda}_k\|^2]$$

$$= \frac{D^2}{2K\eta_k} + \frac{1}{K}\sum_{k=1}^{K}\frac{\eta_k}{2}\mathbb{E}_{\boldsymbol{\xi}}[\|\nabla \boldsymbol{f}(\boldsymbol{x}_k, \boldsymbol{\xi})^T \boldsymbol{\lambda}_k\|^2]$$

$$\leqslant \frac{D^2}{2K\eta_k} + \frac{\sigma_2^2 + G^2}{K}\sum_{k=1}^{K}\eta_k$$

where the last inequality is by Lemma 2. □

**Lemma 4.** *Under the same assumptions as in Theorem 1, we have the following inequality*

$$\mathbb{E}_{\boldsymbol{\xi}}[\frac{1}{K}\sum_{k=1}^{K}(\boldsymbol{\lambda}^* - \boldsymbol{\lambda}_k)^T \boldsymbol{f}(\boldsymbol{x}_k, \boldsymbol{\xi})] \leqslant \frac{\sigma_2^2 + G^2}{K}\sum_{k=1}^{K}\alpha_k + \frac{\sqrt{2}}{K}\sum_{k=1}^{K}\beta_k.$$

*Proof.* From the update rule of $\boldsymbol{\lambda}_k$, where $\beta_k \geqslant 0$ in convex settings, we have

$$-\boldsymbol{\lambda}_k^T \boldsymbol{f}(\boldsymbol{x}_k, \boldsymbol{\xi}) + \frac{\alpha_k}{2}\|\nabla \boldsymbol{f}(\boldsymbol{x}_k, \boldsymbol{\xi})^T \boldsymbol{\lambda}_k\|^2 + \beta_k\|\boldsymbol{\lambda}_k - \boldsymbol{\lambda}_{k-1}\|^2$$

$$\leqslant -\boldsymbol{\lambda}^{*T} \boldsymbol{f}(\boldsymbol{x}_k, \boldsymbol{\xi}) + \frac{\alpha_k}{2}\|\nabla \boldsymbol{f}(\boldsymbol{x}_k, \boldsymbol{\xi})^T \boldsymbol{\lambda}^*\|^2 + \beta_k\|\boldsymbol{\lambda}^* - \boldsymbol{\lambda}_{k-1}\|^2.$$

Since $\frac{\alpha_k}{2}\|\nabla \boldsymbol{f}(\boldsymbol{x}_k, \boldsymbol{\xi})^T \boldsymbol{\lambda}_k\|^2 + \beta_k\|\boldsymbol{\lambda}_k - \boldsymbol{\lambda}_{k-1}\|^2 \geqslant 0$, then

$$-\boldsymbol{\lambda}_k^T \boldsymbol{f}(\boldsymbol{x}_k, \boldsymbol{\xi}) \leqslant -\boldsymbol{\lambda}^{*T} \boldsymbol{f}(\boldsymbol{x}_k, \boldsymbol{\xi}) + \frac{\alpha_k}{2}\|\nabla \boldsymbol{f}(\boldsymbol{x}_k, \boldsymbol{\xi})^T \boldsymbol{\lambda}^*\|^2 + \beta_k\|\boldsymbol{\lambda}^* - \boldsymbol{\lambda}_{k-1}\|^2.$$

Rearranging the above terms, it follows that,

$$(\boldsymbol{\lambda}^* - \boldsymbol{\lambda}_k)^T \boldsymbol{f}(\boldsymbol{x}_k, \boldsymbol{\xi}) \leqslant \frac{\alpha_k}{2}\|\nabla \boldsymbol{f}(\boldsymbol{x}_k, \boldsymbol{\xi})^T \boldsymbol{\lambda}^*\|^2 + \beta_k\|\boldsymbol{\lambda}^* - \boldsymbol{\lambda}_{k-1}\|^2 \leqslant (\sigma_2^2 + G^2)\alpha_k + \sqrt{2}\beta_k$$

Then, we have

$$\mathbb{E}_{\boldsymbol{\xi}}[\frac{1}{K}\sum_{k=1}^{K}(\boldsymbol{\lambda}^* - \boldsymbol{\lambda}_k)^T \boldsymbol{f}(\boldsymbol{x}_k, \boldsymbol{\xi})] \leqslant \mathbb{E}_{\boldsymbol{\xi}}[\frac{1}{K}\sum_{k=1}^{K}\frac{\alpha_k}{2}\|\nabla \boldsymbol{f}(\boldsymbol{x}_k, \boldsymbol{\xi})^T \boldsymbol{\lambda}^*\|^2 + \beta_k\|\boldsymbol{\lambda}^* - \boldsymbol{\lambda}_{k-1}\|^2]$$

$$= \frac{1}{K}\sum_{k=1}^{K}\mathbb{E}_{\boldsymbol{\xi}}[\frac{\alpha_k}{2}\|\nabla \boldsymbol{f}(\boldsymbol{x}_k, \boldsymbol{\xi})^T \boldsymbol{\lambda}^*\|^2] + \frac{1}{K}\sum_{k=1}^{K}\mathbb{E}_{\boldsymbol{\xi}}[\beta_k\|\boldsymbol{\lambda}^* - \boldsymbol{\lambda}_{k-1}\|^2]$$

$$\leqslant \frac{\sigma_2^2 + G^2}{K}\sum_{k=1}^{K}\alpha_k + \frac{\sqrt{2}}{K}\sum_{k=1}^{K}\beta_k.$$

□

Next, we prove Theorem 1.

*Proof.* Since the function $\mathcal{L}(\boldsymbol{x}, \boldsymbol{\lambda})$ is convex with respect to the variable $\boldsymbol{x}$ and concave with respect to the variable $\boldsymbol{\lambda}$, it follows that

$$
\begin{aligned}
&\mathcal{L}(\boldsymbol{x}_k, \boldsymbol{\lambda}_k) - \mathcal{L}(\boldsymbol{x}^*, \boldsymbol{\lambda}_k) \\
&\leqslant \nabla_{\boldsymbol{x}} \mathcal{L}(\boldsymbol{x}_k, \boldsymbol{\lambda}_k)^T (\boldsymbol{x}_k - \boldsymbol{x}^*) \\
&= (\nabla \boldsymbol{f}(\boldsymbol{x}_k)^T \boldsymbol{\lambda}_k)^T (\boldsymbol{x}_k - \boldsymbol{x}^*) \\
&= (\nabla \boldsymbol{f}(\boldsymbol{x}_k, \boldsymbol{\xi})^T \boldsymbol{\lambda}_k)^T (\boldsymbol{x}_k - \boldsymbol{x}^*) + (\nabla \boldsymbol{f}(\boldsymbol{x}_k)^T \boldsymbol{\lambda}_k - \nabla \boldsymbol{f}(\boldsymbol{x}_k, \boldsymbol{\xi})^T \boldsymbol{\lambda}_k)^T (\boldsymbol{x}_k - \boldsymbol{x}^*) \qquad (9)
\end{aligned}
$$

and

$$
\begin{aligned}
\mathcal{L}(\boldsymbol{x}_k, \boldsymbol{\lambda}^*) - \mathcal{L}(\boldsymbol{x}_k, \boldsymbol{\lambda}_k) &\leqslant \nabla_{\boldsymbol{\lambda}} \mathcal{L}(\boldsymbol{x}_k, \boldsymbol{\lambda}_k)^T (\boldsymbol{\lambda}^* - \boldsymbol{\lambda}_k) \\
&= (\boldsymbol{\lambda}^* - \boldsymbol{\lambda}_k)^T \boldsymbol{f}(\boldsymbol{x}_k) \\
&= (\boldsymbol{\lambda}^* - \boldsymbol{\lambda}_k)^T \boldsymbol{f}(\boldsymbol{x}_k, \boldsymbol{\xi}) + (\boldsymbol{\lambda}^* - \boldsymbol{\lambda}_k)^T (\boldsymbol{f}(\boldsymbol{x}_k) - \boldsymbol{f}(\boldsymbol{x}_k, \boldsymbol{\xi})) \qquad (10)
\end{aligned}
$$

Combining the inequalities (9) and (10), we have

$$
\mathbb{E}_{\boldsymbol{\xi}}[\frac{1}{K} \sum_{k=1}^{K} \mathcal{L}(\boldsymbol{x}_k, \boldsymbol{\lambda}^*) - \mathcal{L}(\boldsymbol{x}^*, \boldsymbol{\lambda}_k)]
$$

$$
\leqslant \mathbb{E}_{\boldsymbol{\xi}}[\frac{1}{K} \sum_{k=1}^{K} (\nabla \boldsymbol{f}(\boldsymbol{x}_k, \boldsymbol{\xi})^T \boldsymbol{\lambda}_k)^T (\boldsymbol{x}_k - \boldsymbol{x}^*) + (\boldsymbol{\lambda}^* - \boldsymbol{\lambda}_k)^T \boldsymbol{f}(\boldsymbol{x}_k, \boldsymbol{\xi})]
$$

$$
+ \mathbb{E}_{\boldsymbol{\xi}}[\frac{1}{K} \sum_{k=1}^{K} (\nabla \boldsymbol{f}(\boldsymbol{x}_k)^T \boldsymbol{\lambda}_k - \nabla \boldsymbol{f}(\boldsymbol{x}_k, \boldsymbol{\xi})^T \boldsymbol{\lambda}_k)^T (\boldsymbol{x}_k - \boldsymbol{x}^*) + (\boldsymbol{f}(\boldsymbol{x}_k) - \boldsymbol{f}(\boldsymbol{x}_k, \boldsymbol{\xi}))^T (\boldsymbol{\lambda}^* - \boldsymbol{\lambda}_k)]
$$

$$
\leqslant \frac{D^2}{2K\eta_k} + \frac{\sigma_2^2 + G^2}{K} \sum_{k=1}^{K} \eta_k + \frac{\sigma_2^2 + G^2}{K} \sum_{k=1}^{K} \alpha_k + \frac{\sqrt{2}}{K} \sum_{k=1}^{K} \beta_k
$$

$$
+ \mathbb{E}_{\boldsymbol{\xi}}[\frac{1}{K} \sum_{k=1}^{K} (\mathbb{E}[\boldsymbol{\lambda}_k^T \nabla \boldsymbol{f}(\boldsymbol{x}_k, \boldsymbol{\xi})] - \boldsymbol{\lambda}_k^T \nabla \boldsymbol{f}(\boldsymbol{x}_k, \boldsymbol{\xi}))^T (\boldsymbol{x}_k - \boldsymbol{x}^*)
$$

$$
+ (\mathbb{E}[\boldsymbol{f}(\boldsymbol{x}_k, \boldsymbol{\xi})] - \boldsymbol{f}(\boldsymbol{x}_k, \boldsymbol{\xi}))^T (\boldsymbol{\lambda}^* - \boldsymbol{\lambda}_k)]
$$

where the last inequality is by Lemmas 3 and 4.

Let

$$
H := \sum_{t=1}^{T} (\mathbb{E}[\boldsymbol{\lambda}_k^T \nabla \boldsymbol{f}(\boldsymbol{x}_k, \boldsymbol{\xi})] - \boldsymbol{\lambda}_k^T \nabla \boldsymbol{f}(\boldsymbol{x}_k, \boldsymbol{\xi}))^T (\boldsymbol{x}_k - \boldsymbol{x}^*) + (\mathbb{E}[\boldsymbol{f}(\boldsymbol{x}_k, \boldsymbol{\xi})] - \boldsymbol{f}(\boldsymbol{x}_k, \boldsymbol{\xi}))^T (\boldsymbol{\lambda}^* - \boldsymbol{\lambda}_k).
$$

and $h_f := 2\sqrt{2K(D^2 G^2 + F^2) \ln \frac{1}{\delta}}$, then by Hoeffiding inequality:

$$
P(H \geqslant h_f) \leqslant \delta
$$

Therefore, the following inequality holds with probability $(1 - \delta)$:

$$
\mathbb{E}_{\boldsymbol{\xi}}[\frac{1}{K} \sum_{k=1}^{K} \mathcal{L}(\boldsymbol{x}_k, \boldsymbol{\lambda}^*) - \mathcal{L}(\boldsymbol{x}^*, \boldsymbol{\lambda}_k)] \leqslant \frac{D^2}{2K\eta_k} + \frac{\sigma_2^2 + G^2}{K} \sum_{k=1}^{K} (\eta_k + \alpha_k) + \frac{\sqrt{2}}{K} \sum_{k=1}^{K} \beta_k \qquad (11)
$$

$$
+ \frac{2\sqrt{2(D^2 G^2 + F^2) \ln \frac{1}{\delta}}}{\sqrt{K}}
$$

Further,

$$
\begin{aligned}
\mathbb{E}_{\boldsymbol{\xi}}[\frac{1}{K}\sum_{k=1}^{K}\mathcal{L}(\boldsymbol{x}_k,\boldsymbol{\lambda}^*) - \mathcal{L}(\boldsymbol{x}^*,\boldsymbol{\lambda}_k)] =& \mathbb{E}_{\boldsymbol{\xi}}[\frac{1}{K}\sum_{k=1}^{K}\boldsymbol{\lambda}^{*T}\boldsymbol{f}(\boldsymbol{x}_k) - \boldsymbol{\lambda}_k^T\boldsymbol{f}(\boldsymbol{x}^*)] \\
\geqslant& \mathbb{E}_{\boldsymbol{\xi}}[\frac{1}{K}\sum_{k=1}^{K}\boldsymbol{\lambda}^{*T}\boldsymbol{f}(\boldsymbol{x}_k) - \boldsymbol{\lambda}^{*T}\boldsymbol{f}(\boldsymbol{x}^*)] \\
=& \mathbb{E}_{\boldsymbol{\xi}}[\frac{1}{K}\sum_{k=1}^{K}\boldsymbol{\lambda}^{*T}(\boldsymbol{f}(\boldsymbol{x}_k) - \boldsymbol{f}(\boldsymbol{x}^*))] \\
=& \frac{1}{K}\sum_{k=1}^{K}\mathbb{E}_{\boldsymbol{\xi}}[\boldsymbol{\lambda}^{*T}(\boldsymbol{f}(\boldsymbol{x}_k) - \boldsymbol{f}(\boldsymbol{x}^*))] \quad (12)
\end{aligned}
$$

Combining (11) and (12), the following inequality holds with probability $(1-\delta)$:

$$
\begin{aligned}
\frac{1}{K}\sum_{k=1}^{K}\mathbb{E}_{\boldsymbol{\xi}}[\boldsymbol{\lambda}^{*T}(\boldsymbol{f}(\boldsymbol{x}_k) - \boldsymbol{f}(\boldsymbol{x}^*))] \leqslant& \frac{D^2}{2K\eta_k} + \frac{\sigma_2^2 + G^2}{K}\sum_{k=1}^{K}(\eta_k + \alpha_k) + \frac{\sqrt{2}}{K}\sum_{k=1}^{K}\beta_k \\
& + \frac{2\sqrt{2(D^2G^2 + F^2)\ln\frac{1}{\delta}}}{\sqrt{K}}
\end{aligned}
$$

$\square$

## C.2 PROOF OF THEOREM 2

Before proving Theorem 2, we introduce the following lemmas.

**Lemma 5.** *Under the same assumptions as in Theorem 2, we have the following inequality*

$$
\mathbb{E}_{\boldsymbol{\xi}}[\|\boldsymbol{\lambda}_k - \boldsymbol{\lambda}_{k-1}\|^2] \leqslant \frac{\sqrt{2}(\sigma_1 + F)}{\beta_k} + \frac{\alpha_k(\sigma_2^2 + G^2)}{\beta_k}
$$

*Proof.* From the update rule of $\boldsymbol{\lambda}_k$, where $\beta_k > \frac{1}{2}$ in non-convex settings, we have

$$
\begin{aligned}
& -\boldsymbol{\lambda}_k^T\boldsymbol{f}(\boldsymbol{x}_k,\boldsymbol{\xi}) + \frac{\alpha_k}{2}\|\nabla\boldsymbol{f}(\boldsymbol{x}_k,\boldsymbol{\xi})^T\boldsymbol{\lambda}_k\|^2 + \beta_k\|\boldsymbol{\lambda}_k - \boldsymbol{\lambda}_{k-1}\|^2 \\
& \leqslant -\boldsymbol{\lambda}_{k-1}^T\boldsymbol{f}(\boldsymbol{x}_k,\boldsymbol{\xi}) + \frac{\alpha_k}{2}\|\nabla\boldsymbol{f}(\boldsymbol{x}_k,\boldsymbol{\xi})^T\boldsymbol{\lambda}_{k-1}\|
\end{aligned}
$$

Rearranging the above terms, it follows that,

$$
\begin{aligned}
\mathbb{E}_{\boldsymbol{\xi}}[\beta_k\|\boldsymbol{\lambda}_k - \boldsymbol{\lambda}_{k-1}\|^2] \leqslant& \mathbb{E}_{\boldsymbol{\xi}}[(\boldsymbol{\lambda}_k - \boldsymbol{\lambda}_{k-1})^T\boldsymbol{f}(\boldsymbol{x}_k,\boldsymbol{\xi}) + \frac{\alpha_k}{2}\|\nabla\boldsymbol{f}(\boldsymbol{x}_k,\boldsymbol{\xi})^T\boldsymbol{\lambda}_{k-1}\|^2 \\
& - \frac{\alpha_k}{2}\|\nabla\boldsymbol{f}(\boldsymbol{x}_k,\boldsymbol{\xi})^T\boldsymbol{\lambda}_k\|^2] \\
\leqslant& \underbrace{\mathbb{E}_{\boldsymbol{\xi}}[(\boldsymbol{\lambda}_k - \boldsymbol{\lambda}_{k-1})^T\boldsymbol{f}(\boldsymbol{x}_k,\boldsymbol{\xi})]}_{\text{Part 1}} + \underbrace{\mathbb{E}_{\boldsymbol{\xi}}[\frac{\alpha_k}{2}\|\nabla\boldsymbol{f}(\boldsymbol{x}_k,\boldsymbol{\xi})^T\boldsymbol{\lambda}_{k-1}\|^2]}_{\text{Part 2}} \quad (13)
\end{aligned}
$$

For Part 1 in (13), we have

$$
\begin{aligned}
\mathbb{E}_{\boldsymbol{\xi}}[(\boldsymbol{\lambda}_k - \boldsymbol{\lambda}_{k-1})^T\boldsymbol{f}(\boldsymbol{x}_k,\boldsymbol{\xi})] =& \mathbb{E}_{\boldsymbol{\xi}}[(\boldsymbol{\lambda}_k - \boldsymbol{\lambda}_{k-1})^T(\boldsymbol{f}(\boldsymbol{x}_k,\boldsymbol{\xi}) - \boldsymbol{f}(\boldsymbol{x}_k) + \boldsymbol{f}(\boldsymbol{x}_k))] \\
\leqslant& \mathbb{E}_{\boldsymbol{\xi}}[\|(\boldsymbol{\lambda}_k - \boldsymbol{\lambda}_{k-1})^T(\boldsymbol{f}(\boldsymbol{x}_k,\boldsymbol{\xi}) - \boldsymbol{f}(\boldsymbol{x}_k))\| \\
& + \|(\boldsymbol{\lambda}_k - \boldsymbol{\lambda}_{k-1})^T\boldsymbol{f}(\boldsymbol{x}_k)\|)] \\
\leqslant& \sqrt{2}(\sigma_1 + F) \quad (14)
\end{aligned}
$$

For Part 2 in (13), we have

$$\mathbb{E}_{\boldsymbol{\xi}}[\frac{\alpha_k}{2}\|\nabla\boldsymbol{f}(\boldsymbol{x}_k,\boldsymbol{\xi})^T\boldsymbol{\lambda}_{k-1}\|^2] = \frac{\alpha_k}{2}\mathbb{E}_{\boldsymbol{\xi}}[\|\nabla\boldsymbol{f}(\boldsymbol{x}_k,\boldsymbol{\xi})^T\boldsymbol{\lambda}_{k-1} - \nabla\boldsymbol{f}(\boldsymbol{x}_k)^T\boldsymbol{\lambda}_{k-1} + \nabla\boldsymbol{f}(\boldsymbol{x}_k)^T\boldsymbol{\lambda}_{k-1}\|^2]$$
$$\leqslant \frac{\alpha_k}{2}\mathbb{E}_{\boldsymbol{\xi}}[2\|\nabla\boldsymbol{f}(\boldsymbol{x}_k,\boldsymbol{\xi})^T\boldsymbol{\lambda}_{k-1} - \nabla\boldsymbol{f}(\boldsymbol{x}_k)^T\boldsymbol{\lambda}_{k-1}\|^2$$
$$+ 2\|\nabla\boldsymbol{f}(\boldsymbol{x}_k)^T\boldsymbol{\lambda}_{k-1}\|^2]$$
$$\leqslant \alpha_k(\sigma_2^2 + G^2) \tag{15}$$

From (14) and (15), then for (13) we have

$$\mathbb{E}_{\boldsymbol{\xi}}[\beta_k\|\boldsymbol{\lambda}_k - \boldsymbol{\lambda}_{k-1}\|^2] \leqslant \sqrt{2}(\sigma_1 + F) + \alpha_k(\sigma_2^2 + G^2)$$

Since $\beta_k > 0$ in non-convex settings, then

$$\mathbb{E}_{\boldsymbol{\xi}}[\|\boldsymbol{\lambda}_k - \boldsymbol{\lambda}_{k-1}\|^2] \leqslant \frac{\sqrt{2}(\sigma_1 + F)}{\beta_k} + \frac{\alpha_k(\sigma_2^2 + G^2)}{\beta_k}$$

$\square$

**Lemma 6.** *Under the same assumptions as in Theorem 2, we have the following inequality*

$$\mathbb{E}_{\boldsymbol{\xi}}[(\nabla\boldsymbol{f}(\boldsymbol{x}_k)^T\boldsymbol{\lambda}_k)^T(-\nabla\boldsymbol{f}(\boldsymbol{x}_k,\boldsymbol{\xi})^T\boldsymbol{\lambda}_k)] \leqslant \frac{2pG\sigma_2(\sigma_1 + F)}{\sqrt{\beta_k - 0.5}} + \frac{2pG\sigma_2\sqrt{(\sigma_2^2 + G^2)\alpha_k}}{\sqrt{\beta_k - 0.5}}$$
$$- \mathbb{E}_{\boldsymbol{\xi}}[\|\nabla\boldsymbol{f}(\boldsymbol{x}_k)^T\boldsymbol{\lambda}_k\|^2]$$

*Proof.*

$$\mathbb{E}_{\boldsymbol{\xi}}[(\nabla\boldsymbol{f}(\boldsymbol{x}_k)^T\boldsymbol{\lambda}_k)^T(-\nabla\boldsymbol{f}(\boldsymbol{x}_k,\boldsymbol{\xi})^T\boldsymbol{\lambda}_k)]$$
$$= \mathbb{E}_{\boldsymbol{\xi}}[(\nabla\boldsymbol{f}(\boldsymbol{x}_k)^T\boldsymbol{\lambda}_k)^T(\nabla\boldsymbol{f}(\boldsymbol{x}_k)^T\boldsymbol{\lambda}_k - \nabla\boldsymbol{f}(\boldsymbol{x}_k,\boldsymbol{\xi})^T\boldsymbol{\lambda}_k - \nabla\boldsymbol{f}(\boldsymbol{x}_k)^T\boldsymbol{\lambda}_k)]$$
$$= \mathbb{E}_{\boldsymbol{\xi}}[(\nabla\boldsymbol{f}(\boldsymbol{x}_k)^T\boldsymbol{\lambda}_k)^T(\nabla\boldsymbol{f}(\boldsymbol{x}_k)^T\boldsymbol{\lambda}_k - \nabla\boldsymbol{f}(\boldsymbol{x}_k,\boldsymbol{\xi})^T\boldsymbol{\lambda}_k)] - \mathbb{E}_{\boldsymbol{\xi}}[\|\nabla\boldsymbol{f}(\boldsymbol{x}_k)^T\boldsymbol{\lambda}_k\|^2] \tag{16}$$

For the first term of (16),

$$\mathbb{E}_{\boldsymbol{\xi}}[(\nabla\boldsymbol{f}(\boldsymbol{x}_k)^T\boldsymbol{\lambda}_k)^T(\nabla\boldsymbol{f}(\boldsymbol{x}_k)^T\boldsymbol{\lambda}_k - \nabla\boldsymbol{f}(\boldsymbol{x}_k,\boldsymbol{\xi})^T\boldsymbol{\lambda}_k)]$$
$$= \mathbb{E}_{\boldsymbol{\xi}}[\langle\nabla\boldsymbol{f}(\boldsymbol{x}_k)^T\boldsymbol{\lambda}_k, (\nabla\boldsymbol{f}(\boldsymbol{x}_k) - \nabla\boldsymbol{f}(\boldsymbol{x}_k,\boldsymbol{\xi})^T\boldsymbol{\lambda}_k\rangle]$$
$$= \mathbb{E}_{\boldsymbol{\xi}}[\langle\nabla\boldsymbol{f}(\boldsymbol{x}_k)^T\boldsymbol{\lambda}_k, (\nabla\boldsymbol{f}(\boldsymbol{x}_k) - \nabla\boldsymbol{f}(\boldsymbol{x}_k,\boldsymbol{\xi})^T(\boldsymbol{\lambda}_k - \boldsymbol{\lambda}_{k+1})\rangle]$$
$$+ \mathbb{E}_{\boldsymbol{\xi}}[\langle\nabla\boldsymbol{f}(\boldsymbol{x}_k)^T(\boldsymbol{\lambda}_k - \boldsymbol{\lambda}_{k+1}), (\nabla\boldsymbol{f}(\boldsymbol{x}_k) - \nabla\boldsymbol{f}(\boldsymbol{x}_k,\boldsymbol{\xi}))^T\boldsymbol{\lambda}_{k+1}\rangle]$$
$$+ \mathbb{E}_{\boldsymbol{\xi}}[\langle\nabla\boldsymbol{f}(\boldsymbol{x}_k)^T\boldsymbol{\lambda}_{k+1}, (\nabla\boldsymbol{f}(\boldsymbol{x}_k) - \nabla\boldsymbol{f}(\boldsymbol{x}_k,\boldsymbol{\xi}))^T\boldsymbol{\lambda}_{k+1}\rangle]$$
$$\leqslant \mathbb{E}_{\boldsymbol{\xi}}[\|\nabla\boldsymbol{f}(\boldsymbol{x}_k)^T\boldsymbol{\lambda}_k\|\|\boldsymbol{\lambda}_k - \boldsymbol{\lambda}_{k+1}\|\|\nabla\boldsymbol{f}(\boldsymbol{x}_k) - \nabla\boldsymbol{f}(\boldsymbol{x}_k,\boldsymbol{\xi})\|]$$
$$+ \mathbb{E}_{\boldsymbol{\xi}}[\|\boldsymbol{\lambda}_k - \boldsymbol{\lambda}_{k+1}\|\|\nabla\boldsymbol{f}(\boldsymbol{x}_k)\|\|\boldsymbol{\lambda}_{k+1}\|\|\nabla\boldsymbol{f}(\boldsymbol{x}_k) - \nabla\boldsymbol{f}(\boldsymbol{x}_k,\boldsymbol{\xi})\|]$$
$$+ \langle\nabla\boldsymbol{f}(\boldsymbol{x}_k)^T\boldsymbol{\lambda}_{k+1}, (\nabla\boldsymbol{f}(\boldsymbol{x}_k) - \mathbb{E}_{\boldsymbol{\xi}}[\nabla\boldsymbol{f}(\boldsymbol{x}_k,\boldsymbol{\xi})])^T\boldsymbol{\lambda}_{k+1}\rangle$$
$$\leqslant 2pG\mathbb{E}_{\boldsymbol{\xi}}[\|\boldsymbol{\lambda}_k - \boldsymbol{\lambda}_{k-1}\|\|\nabla\boldsymbol{f}(\boldsymbol{x}_k) - \nabla\boldsymbol{f}(\boldsymbol{x}_k,\boldsymbol{\xi})\|]$$
$$\leqslant 2pG\mathbb{E}_{\boldsymbol{\xi}}[\frac{\gamma}{2}\|\boldsymbol{\lambda}_k - \boldsymbol{\lambda}_{k-1}\|^2 + \frac{1}{2\gamma}\|\nabla\boldsymbol{f}(\boldsymbol{x}_k) - \nabla\boldsymbol{f}(\boldsymbol{x}_k,\boldsymbol{\xi})\|^2]$$
$$\leqslant 2pG[\frac{\gamma}{2}(\frac{\sqrt{2}(\sigma_1 + F)}{\beta_k} + \frac{\alpha_k(\sigma_2^2 + G^2)}{\beta_k}) + \frac{\sigma_2^2}{2\gamma}]$$

where the last inequality is by Lemma 5. Let $C_2 = \frac{\sqrt{2}(\sigma_1+F)}{\beta_k} + \frac{\alpha_k(\sigma_2^2+G^2)}{\beta_k}$ and $\gamma = \frac{\sigma_2}{\sqrt{C_2}}$, then

$$\mathbb{E}_{\boldsymbol{\xi}}[(\nabla\boldsymbol{f}(\boldsymbol{x}_k)^T\boldsymbol{\lambda}_k)^T(\nabla\boldsymbol{f}(\boldsymbol{x}_k)^T\boldsymbol{\lambda}_k - \nabla\boldsymbol{f}(\boldsymbol{x}_k,\boldsymbol{\xi})^T\boldsymbol{\lambda}_k)]$$
$$\leqslant 2pG\sigma_2\sqrt{C_2}$$
$$\leqslant 2pG\sigma_2(\frac{\sqrt{2(\sigma_1 + F)}}{\sqrt{\beta_k}} + \frac{\sqrt{\alpha_k(\sigma_2^2 + G^2)}}{\sqrt{\beta_k}}).$$

Thus,

$$\mathbb{E}_{\boldsymbol{\xi}}[(\nabla\boldsymbol{f}(\boldsymbol{x}_k)^T\boldsymbol{\lambda}_k)^T(-\nabla\boldsymbol{f}(\boldsymbol{x}_k,\boldsymbol{\xi})^T\boldsymbol{\lambda}_k)]$$

$$\leqslant 2pG\sigma_2(\frac{\sqrt{2(\sigma_1+F)}}{\sqrt{\beta_k}}+\frac{\sqrt{\alpha_k(\sigma_2^2+G^2)}}{\sqrt{\beta_k}})-\mathbb{E}_{\boldsymbol{\xi}}[\|\nabla\boldsymbol{f}(\boldsymbol{x}_k)^T\boldsymbol{\lambda}_k\|^2]$$

$$\leqslant\frac{2pG\sigma_2\sqrt{2(\sigma_1+F)}}{\sqrt{\beta_k}}+\frac{2pG\sigma_2\sqrt{(\sigma_2^2+G^2)\alpha_k}}{\sqrt{\beta_k}}-\mathbb{E}_{\boldsymbol{\xi}}[\|\nabla\boldsymbol{f}(\boldsymbol{x}_k)^T\boldsymbol{\lambda}_k\|^2]$$

$\square$

Next, we prove Theorem 2.

*Proof.* Since the $L$-smoothness of each objective function, then for any $i \in [p]$,

$$\boldsymbol{\lambda}_k^i\boldsymbol{f}^i(\boldsymbol{x}_{k+1}) \leqslant \boldsymbol{\lambda}_k^i\boldsymbol{f}^i(\boldsymbol{x}_k) + (\nabla\boldsymbol{f}^i(\boldsymbol{x}_k)\boldsymbol{\lambda}_k^i)^T(\boldsymbol{x}_{k+1}-\boldsymbol{x}_k) + \frac{\boldsymbol{\lambda}_k^i L}{2}\|\boldsymbol{x}_{k+1}-\boldsymbol{x}_k\|^2$$

$$=\boldsymbol{\lambda}_k^i\boldsymbol{f}^i(\boldsymbol{x}_k) + (\nabla\boldsymbol{f}^i(\boldsymbol{x}_k)\boldsymbol{\lambda}_k^i)^T(-\eta_k\nabla\boldsymbol{f}(\boldsymbol{x}_k,\boldsymbol{\xi})^T\boldsymbol{\lambda}_k) + \frac{\boldsymbol{\lambda}_k^i L\eta_k^2}{2}\|\nabla\boldsymbol{f}(\boldsymbol{x}_k,\boldsymbol{\xi})^T\boldsymbol{\lambda}_k\|^2$$

Sum up both side for $i = 1, \cdots, p$,

$$\boldsymbol{\lambda}_k^T\boldsymbol{f}(\boldsymbol{x}_{k+1}) \leqslant \boldsymbol{\lambda}_k^T\boldsymbol{f}(\boldsymbol{x}_k) + (\nabla\boldsymbol{f}(\boldsymbol{x}_k)^T\boldsymbol{\lambda}_k)^T(-\eta_k\nabla\boldsymbol{f}(\boldsymbol{x}_k,\boldsymbol{\xi})^T\boldsymbol{\lambda}_k) + \frac{L\eta_k^2}{2}\|\nabla\boldsymbol{f}(\boldsymbol{x}_k,\boldsymbol{\xi})^T\boldsymbol{\lambda}_k\|^2$$

For the above inequality, take the expectation on random variable $\boldsymbol{\xi}$,

$$\mathbb{E}_{\boldsymbol{\xi}}[\boldsymbol{\lambda}_k^T\boldsymbol{f}(\boldsymbol{x}_{k+1})] \leqslant \mathbb{E}_{\boldsymbol{\xi}}[\boldsymbol{\lambda}_k^T\boldsymbol{f}(\boldsymbol{x}_k)] + \eta_k\mathbb{E}_{\boldsymbol{\xi}}[(\nabla\boldsymbol{f}(\boldsymbol{x}_k)^T\boldsymbol{\lambda}_k)^T(-\nabla\boldsymbol{f}(\boldsymbol{x}_k,\boldsymbol{\xi})^T\boldsymbol{\lambda}_k)]$$

$$+ \mathbb{E}_{\boldsymbol{\xi}}[\frac{L\eta_k^2}{2}\|\nabla\boldsymbol{f}(\boldsymbol{x}_k,\boldsymbol{\xi})^T\boldsymbol{\lambda}_k\|^2]$$

$$\leqslant\mathbb{E}_{\boldsymbol{\xi}}[\boldsymbol{\lambda}_k^T\boldsymbol{f}(\boldsymbol{x}_k)] + \eta_k(\frac{2pG\sigma_2\sqrt{2(\sigma_1+F)}}{\sqrt{\beta_k}}+\frac{2pG\sigma_2\sqrt{(\sigma_2^2+G^2)\alpha_k}}{\sqrt{\beta_k}})$$

$$- \eta_k\mathbb{E}_{\boldsymbol{\xi}}[\|\nabla\boldsymbol{f}(\boldsymbol{x}_k)^T\boldsymbol{\lambda}_k\|^2] + L\eta_k^2(\sigma_2^2+G^2)$$

where the last inequality is by Lemma 6. Rearranging the above terms,

$$\eta_k\mathbb{E}_{\boldsymbol{\xi}}[\|\nabla\boldsymbol{f}(\boldsymbol{x}_k)^T\boldsymbol{\lambda}_k\|^2] \leqslant\mathbb{E}_{\boldsymbol{\xi}}[\boldsymbol{\lambda}_k^T\boldsymbol{f}(\boldsymbol{x}_k) - \boldsymbol{\lambda}_k^T\boldsymbol{f}(\boldsymbol{x}_{k+1})] + L\eta_k^2(\sigma_2^2+G^2)$$

$$+ \eta_k(\frac{2pG\sigma_2\sqrt{2(\sigma_1+F)}}{\sqrt{\beta_k}}+\frac{2pG\sigma_2\sqrt{(\sigma_2^2+G^2)\alpha_k}}{\sqrt{\beta_k}}).$$

setting $\eta_k$ ($\frac{1}{\eta_0^p}:=0$) be monotonically decreasing as $k$ increases, then

$$\mathbb{E}_{\boldsymbol{\xi}}[\|\nabla\boldsymbol{f}(\boldsymbol{x}_k)^T\boldsymbol{\lambda}_k\|^2] \leqslant \frac{1}{\eta_K}\mathbb{E}_{\boldsymbol{\xi}}[\boldsymbol{\lambda}_k^T\boldsymbol{f}(\boldsymbol{x}_k) - \boldsymbol{\lambda}_k^T\boldsymbol{f}(\boldsymbol{x}_{k+1})] + \eta_kL(\sigma_2^2+G^2) \qquad (17)$$

$$+\frac{2pG\sigma_2\sqrt{2(\sigma_1+F)}}{\sqrt{\beta_k}}+\frac{2pG\sigma_2\sqrt{(\sigma_2^2+G^2)\alpha_k}}{\sqrt{\beta_k}}$$

Taking the average of $K$ rounds for (17), we have

$$\frac{1}{K}\sum_{k=1}^{K}\mathbb{E}_{\boldsymbol{\xi}}[\nabla\boldsymbol{f}(\boldsymbol{x}_k)^T\boldsymbol{\lambda}_k\|^2]$$

$$\leqslant\frac{1}{K\eta_K}\sum_{k=1}^{K}\mathbb{E}_{\boldsymbol{\xi}}[\boldsymbol{\lambda}_k^T\boldsymbol{f}(\boldsymbol{x}_k) - \boldsymbol{\lambda}_k^T\boldsymbol{f}(\boldsymbol{x}_{k+1})] + \frac{2pG\sigma_2\sqrt{2(\sigma_1+F)}}{K}\sum_{k=1}^{K}\beta_k^{-1/2}$$

$$+\frac{2pG\sigma_2\sqrt{\sigma_2^2+G^2}}{K}\sum_{k=1}^{K}\alpha_k^{1/2}\beta_k^{-1/2} + \frac{L(\sigma_2^2+G^2)}{K}\sum_{k=1}^{K}\eta_k \qquad (18)$$

Furthermore, for the first term on the right-hand side in (18),

$$
\begin{aligned}
\sum_{k=1}^{K} \mathbb{E}_{\boldsymbol{\xi}}[\boldsymbol{\lambda}_k^T \boldsymbol{f}(\boldsymbol{x}_k) - \boldsymbol{\lambda}_k^T \boldsymbol{f}(\boldsymbol{x}_{k+1})] =& \mathbb{E}_{\boldsymbol{\xi}}[\sum_{k=1}^{K} \boldsymbol{\lambda}_k^T \boldsymbol{f}(\boldsymbol{x}_k) - \boldsymbol{\lambda}_k^T \boldsymbol{f}(\boldsymbol{x}_{k+1})] \\
=& \mathbb{E}_{\boldsymbol{\xi}}[\sum_{k=2}^{K} (\boldsymbol{\lambda}_k - \boldsymbol{\lambda}_{k-1})^T \boldsymbol{f}(\boldsymbol{x}_k) + \boldsymbol{\lambda}_1^T \boldsymbol{f}(\boldsymbol{x}_1) - \boldsymbol{\lambda}_K^T \boldsymbol{f}(\boldsymbol{x}_{k+1})] \\
\leqslant& \mathbb{E}_{\boldsymbol{\xi}}[\sum_{k=2}^{K} F\|\boldsymbol{\lambda}_k - \boldsymbol{\lambda}_{k-1}\| + \|\boldsymbol{\lambda}_1^T \boldsymbol{f}(\boldsymbol{x}_1) - \boldsymbol{\lambda}_K^T \boldsymbol{f}(\boldsymbol{x}_{k+1})\|] \\
\leqslant& F \mathbb{E}_{\boldsymbol{\xi}}[\sum_{k=2}^{K} \|\boldsymbol{\lambda}_k - \boldsymbol{\lambda}_{k-1}\|] + 2F \\
\leqslant& F \sum_{k=2}^{K} \sqrt{\frac{\sqrt{2}(\sigma_1 + F)}{\beta_k} + \frac{\alpha_k(\sigma_2^2 + G^2)}{\beta_k}} + 2F \\
\leqslant& F\sqrt{2(\sigma_1 + F)} \sum_{k=2}^{K} \beta_k^{-1/2} + F\sqrt{\sigma_2^2 + G^2} \sum_{k=2}^{K} \alpha_k^{1/2} \beta_k^{-1/2} + 2F
\end{aligned}
$$

where the penultimate inequality is by Lemma 5.

Then, we have

$$
\begin{aligned}
\frac{1}{K} \sum_{k=1}^{K} \mathbb{E}_{\boldsymbol{\xi}}[\|\nabla \boldsymbol{f}(\boldsymbol{x}_k)^T \boldsymbol{\lambda}_k\|^2] \leqslant& \frac{F\sqrt{2(\sigma_1 + F)}}{K\eta_K} \sum_{k=2}^{K} \beta_k^{-1/2} + \frac{F\sqrt{\sigma_2^2 + G^2}}{K\eta_K} \sum_{k=2}^{K} \alpha_k^{1/2} \beta_k^{-1/2} + \frac{2F}{K\eta_K} \\
&+ \frac{2pG\sigma_2 \sqrt{2(\sigma_1 + F)}}{K} \sum_{k=1}^{K} \beta_k^{-1/2} + \frac{2pG\sigma_2 \sqrt{\sigma_2^2 + G^2}}{K} \sum_{k=1}^{K} \alpha_k^{1/2} \beta_k^{-1/2} \\
&+ \frac{L(\sigma_2^2 + G^2)}{K} \sum_{k=1}^{K} \eta_k
\end{aligned}
$$

Since $\frac{1}{K} \sum_{k=1}^{K} \mathbb{E}_{\boldsymbol{\xi}}[\min_{\boldsymbol{\lambda}_k^* \in \Lambda} \|\nabla \boldsymbol{f}(\boldsymbol{x}_k)^T \boldsymbol{\lambda}_k^*\|^2] \leqslant \frac{1}{K} \sum_{k=1}^{K} \mathbb{E}_{\boldsymbol{\xi}}[\|\nabla \boldsymbol{f}(\boldsymbol{x}_k)^T \boldsymbol{\lambda}_k\|^2]$, then Theorem 2 holds. $\square$

## D   MORE EXPERIMENTAL DETAILS AND RESULTS

In this section, we provide more details of the experiment and the experimental results. All experiments are conducted on a single NVIDIA GeForce RTX 4090 GPU, PyTorch 2.1.0 platform.

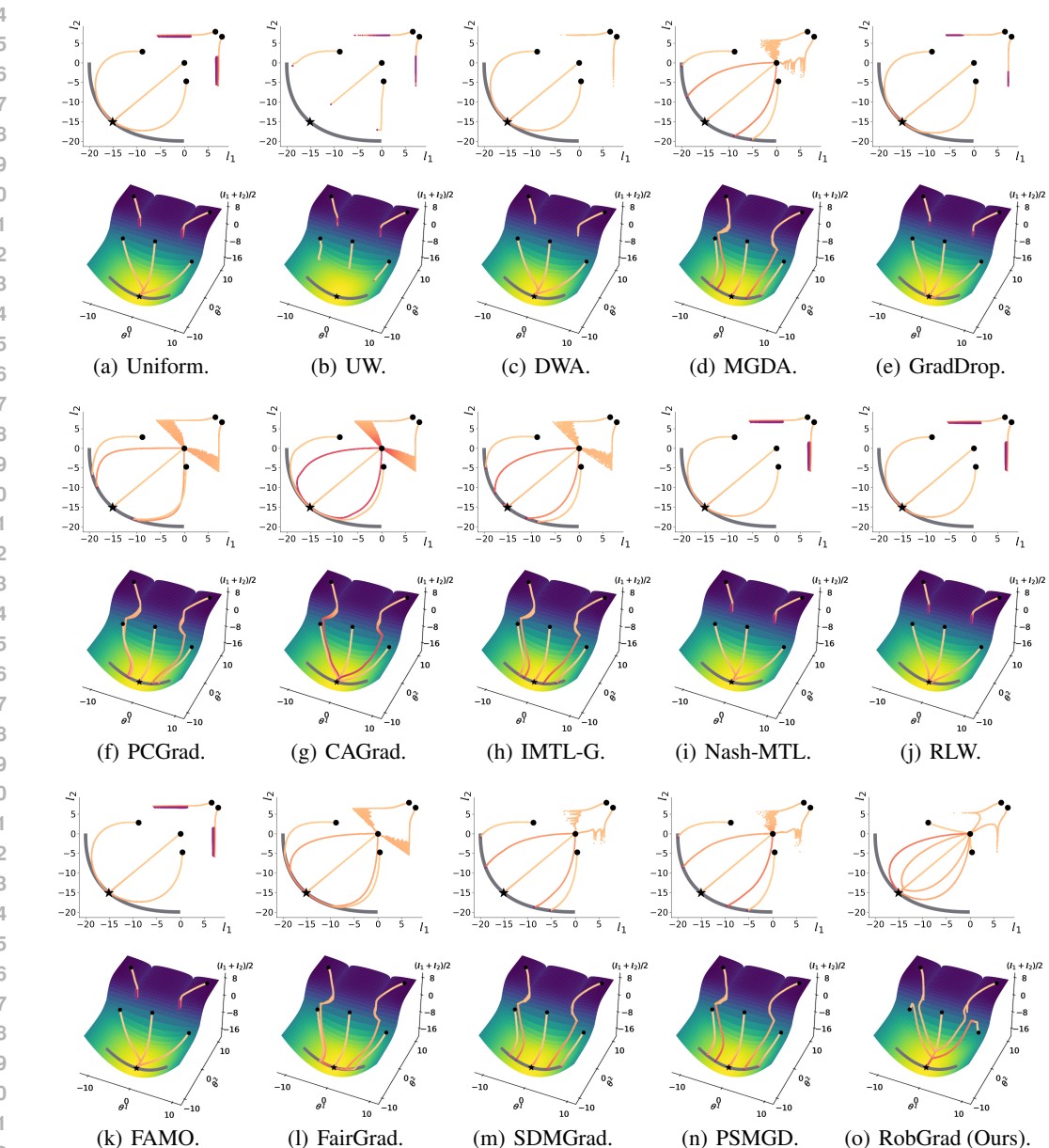

Figure 4: Comparison of SMOO algorithms on a synthetic two-objective problem (Liu et al., 2021a). The optimization trajectories of algorithms based on five different initial points in the objective space ($l_1, l_2$, top row) and the decision-mean objective space (($\theta^1, \theta^2$)-$(l_1 + l_2)/2$, bottom row) are plotted. In figures, the black dots (●) are initial points, the gray curve is the Pareto front, and the black pentagram (★) is the Pareto front point with balanced performance.

## D.1 SYNTHETIC EXAMPLE

The synthetic example is a bi-objective optimization problem as follows (Liu et al., 2021a).

$$\min_{\boldsymbol{\theta}}(l_1(\boldsymbol{\theta}), l_2(\boldsymbol{\theta}))^T = (c_1(\boldsymbol{\theta})f_1(\boldsymbol{\theta}) + c_2(\boldsymbol{\theta})g_1(\boldsymbol{\theta}), c_1(\boldsymbol{\theta})f_2(\boldsymbol{\theta}) + c_2(\boldsymbol{\theta})g_2(\boldsymbol{\theta}))^T,$$

where

$$c_1(\boldsymbol{\theta}) = \max(\tanh(0.5 * \boldsymbol{\theta}^2), 0), \; c_2(\boldsymbol{\theta}) = \max(\tanh(-0.5 * \boldsymbol{\theta}_2), 0)$$
$$f_1(\boldsymbol{\theta}) = \log(\max(|0.5(-\boldsymbol{\theta}^1 - 7) - \tanh(-\boldsymbol{\theta}^2)|, 0.000005)) + 6,$$
$$f_2(\boldsymbol{\theta}) = \log(\max(|0.5(-\boldsymbol{\theta}^1 + 3) - \tanh(-\boldsymbol{\theta}^2 + 2)|, 0.000005)) + 6,$$
$$g_1(\boldsymbol{\theta}) = ((-\boldsymbol{\theta}^1 + 7)^2 + 0.1 * (-\boldsymbol{\theta}^2 - 8)^2)/10 - 20,$$
$$g_2(\boldsymbol{\theta}) = ((-\boldsymbol{\theta}^1 - 7)^2 + 0.1 * (-\boldsymbol{\theta}^2 - 8)^2)/10 - 20.$$

We set five initial solutions: $(-8.5, 7.5)^T$, $(0.0, 0.0)^T$, $(9.0, 9.0)^T$, $(-7.5, -0.5)^T$, $(9.0, -1.0)^T$. We use Adam (Kingma, 2014) optimizer with learning rate $10^{-3}$ and optimize for 50k iterations. Figure 4 plots the results of all the algorithms on this synthetic example. The results show that Uniform, UW, DWA, GradDrop, Nash-MTL, RLW and FAMO all fall into local optimal solutions. MGDA, PCGrad, IMTL-G, SDMGrad and PSMGD are significantly sensitive to the initial solution. For five different initial solutions, CAGrad, FairGrad and RobGrad all converge stably to the Pareto front point with balanced performance. However, as can be seen from their trajectories and their colors, RobGrad is able to escape the local optimum solution more quickly than CAGrad and FairGrad.

### D.2    MORE RESULTS FOR CITYSCAPES, NYU-V2, CELEBA, AND MULTI-MNIST

For the datasets CityScapes and NYU-v2, we follow the experimental setup from Liu et al. (2023). Thus, the metric values for the baseline methods are taken from their original paper, and we provide RobGrad's result with error bars as follows, i.e., Table 4 and 5.

Table 4: The results on the CityScapes dataset (2 tasks). The mean and standard deviation (stdev) of each metric ($\uparrow$ indicates higher better and $\downarrow$ indicates lower better) is reported over 3 independent runs.

| Method | Segmentation | | Depth | | $\Delta M\% \downarrow$ |
|---|---|---|---|---|---|
| | mIoU $\uparrow$ | Pix Acc $\uparrow$ | Abs Err $\downarrow$ | Rel Err $\downarrow$ | |
| RobGrad (mean) | 75.91 | 93.72 | 0.0134 | 32.81 | 5.55 |
| RobGrad (stdev) | $\pm 0.31$ | $\pm 0.03$ | $\pm 0.0009$ | $\pm 0.54$ | $\pm 1.74$ |

Table 5: The results on the NYU-v2 dataset (3 tasks). The mean and standard deviation (stdev) of each metric ($\uparrow$ indicates higher better and $\downarrow$ indicates lower better) is reported over 3 independent runs.

| Method | Segmentation | | Depth | | Surface Normal | | | | | $\Delta M\% \downarrow$ |
|---|---|---|---|---|---|---|---|---|---|---|
| | | | | | Angle Distance $\downarrow$ | | Within $t°$ $\uparrow$ | | | |
| | mIoU $\uparrow$ | Pix Acc $\uparrow$ | Abs Err $\downarrow$ | Rel Err $\downarrow$ | Mean | Median | 11.25 | 22.5 | 30 | |
| RobGrad (mean) | 38.77 | 65.01 | 0.5422 | 0.2235 | 24.79 | 18.89 | 30.71 | 58.08 | 70.03 | -5.53 |
| RobGrad (stdev) | $\pm 0.09$ | $\pm 0.42$ | $\pm 0.0026$ | $\pm 0.0057$ | $\pm 0.14$ | $\pm 0.13$ | $\pm 0.22$ | $\pm 0.22$ | $\pm 0.13$ | $\pm 0.42$ |

For the CelebA, the mean and standard deviation of $\Delta M\%$ for algorithms are reported in Table 6.

Table 6: The results on the CelebA dataset (3 tasks). The mean and standard deviation of $\Delta M\%$ is reported over 3 independent runs.

| Method | Uniform | UW | DWA | MGDA | GradDrop | PCGrad | CAGrad |
|---|---|---|---|---|---|---|---|
| $\Delta M\%$ | $6.46 \pm 1.42$ | $5.68 \pm 0.27$ | $6.27 \pm 0.83$ | $10.15 \pm 2.21$ | $6.99 \pm 0.35$ | $6.87 \pm 1.51$ | $6.09 \pm 1.11$ |
| Method | IMTL-G | Nash-MTL | RLW | FAMO | FairGrad | PSMGD | RobGrad |
| $\Delta M\%$ | $4.61 \pm 0.47$ | $4.52 \pm 1.42$ | $5.55 \pm 0.63$ | $4.26 \pm 0.89$ | $3.25 \pm 1.70$ | $5.11 \pm 0.51$ | $3.09 \pm 0.39$ |

The Multi-MNIST dataset is acquired by the code provided by Sener & Koltun (2018). It is a multi-task version of the MNIST dataset, where each sample consists of an overlap of two digital images from MNIST at the top-left and another one at the bottom-right. The tasks are simultaneously classifying the digit on the top-left (task-L) and the bottom-right (task-R) separately. We follow the setup of Sener & Koltun (2018) and use the LeNet network architecture.

Moreover, we compare the runtime of the algorithms on CityScapes dataset. Table 7 records the mean and standard deviation of the runtime for each algorithm based on over three independent runs. In Table 7, among 15 comparison algorithms, our RobGrad ranks 7th in average runtime, placing it in the upper-middle range of speed. Although RobGrad is not the fastest among all algorithms, it achieves the best overall performance metrics across multiple datasets. In other words, our approach strikes a favorable balance between speed and performance.

Table 7: The average per-epoch runtime of the algorithms on CityScapes dataset. The mean and standard deviation of runtime is reported over 3 independent runs.

| Method | Uniform | UW | DWA | MGDA | GradDrop |
|---|---|---|---|---|---|
| Time $\downarrow$ | $\mathbf{1.05 \pm 0.16}$ | $1.16 \pm 0.18$ | $1.57 \pm 0.32$ | $3.41 \pm 0.21$ | $2.94 \pm 0.17$ |
| Method | PCGrad | CAGrad | IMTL-G | Nash-MTL | RLW |
| Time $\downarrow$ | $3.07 \pm 0.08$ | $2.87 \pm 0.13$ | $2.90 \pm 0.15$ | $3.21 \pm 0.25$ | $1.68 \pm 0.23$ |
| Method | FAMO | FairGrad | SDMGrad | PSMGD | RobGrad |
| Time $\downarrow$ | $1.70 \pm 0.34$ | $2.95 \pm 0.13$ | $11.15 \pm 0.46$ | $2.01 \pm 0.15$ | $2.38 \pm 0.16$ |

### D.3 Additional Multi-task Learning Experiments: QM-9

QM-9 (Blum & Reymond, 2009) is a standardized benchmark dataset in the field of quantum chemistry. This dataset contains the results of quantum chemical calculations for about 134K molecules, which are presented as graphs and are labeled with node and edge features. Among them, 110K molecules were used as a training set, 10K for the validation set, and the remaining 10K for the test set. We follow the same experimental setting used in FAMO (Liu et al., 2023) and PSMGD (Xu et al., 2025). The results are reported in Table 8. The results show that Robgrad achieves new SOTA results on the comprehensive metrics MR and $\Delta M\%$ for the QM-9 dataset.

Table 8: The results on the QM-9 dataset (11 tasks). The mean of each metric ($\uparrow$ indicates higher better and $\downarrow$ indicates lower better) is reported over 3 independent runs. The best average result is marked in bold. MR and $\Delta M\%$ are the main metrics. **MR** and $\mathbf{\Delta M\%}$ are the main metrics.

| Method | $\mu$ | $\alpha$ | $\epsilon_{HOMO}$ | $\epsilon_{LUMO}$ | $\langle R^2 \rangle$ | ZPVE | $U_0$ | $U$ | $H$ | $G$ | $c_v$ | MR $\downarrow$ | $\Delta M\% \downarrow$ |
|---|---|---|---|---|---|---|---|---|---|---|---|---|---|
| | | | | | | MAE $\downarrow$ | | | | | | | |
| Single-task | 0.07 | 0.18 | 60.6 | 53.9 | 0.50 | 4.53 | 58.8 | 64.2 | 63.8 | 66.2 | 0.07 | | |
| Uniform | 0.11 | 0.33 | **73.6** | 89.7 | 5.20 | 14.06 | 143.4 | 144.2 | 144.6 | 140.3 | 0.13 | 9.18 | 177.6 |
| UW | 0.39 | 0.43 | 166.2 | 155.8 | **1.07** | 4.99 | **66.4** | **66.8** | **66.8** | **66.2** | 0.12 | 6.09 | 108.0 |
| DWA | 0.11 | 0.33 | 74.1 | 90.6 | 5.09 | 13.99 | 142.3 | 143.0 | 143.4 | 139.3 | 0.13 | 8.82 | 175.3 |
| MGDA | 0.22 | 0.37 | 126.8 | 104.6 | 3.23 | 5.69 | 88.4 | 89.4 | 89.3 | 88.0 | 0.12 | 8.64 | 120.5 |
| PCGrad | 0.11 | 0.29 | 75.9 | 88.3 | 3.94 | 9.15 | 116.4 | 116.8 | 117.2 | 114.5 | 0.11 | 7.45 | 125.7 |
| CAGrad | 0.12 | 0.32 | 83.5 | 94.8 | 3.22 | 6.93 | 114.0 | 114.3 | 114.5 | 112.3 | 0.12 | 8.55 | 112.8 |
| IMTL-G | 0.14 | 0.29 | 98.3 | 93.9 | 1.75 | 5.70 | 101.4 | 102.4 | 102.0 | 100.1 | 0.10 | 7.00 | 77.2 |
| Nash-MTL | **0.10** | **0.25** | 82.9 | 81.9 | 2.43 | 5.38 | 74.5 | 75.0 | 75.1 | 74.2 | **0.09** | 4.00 | 62.0 |
| RLW | 0.11 | 0.34 | 76.9 | 92.8 | 5.87 | 15.47 | 156.3 | 157.1 | 157.6 | 153.0 | 0.14 | 10.55 | 203.8 |
| FAMO | 0.15 | 0.30 | 94.0 | 95.2 | 1.63 | **4.95** | 70.82 | 71.2 | 71.2 | 70.3 | 0.10 | 5.36 | 58.5 |
| FairGrad | 0.12 | **0.25** | 87.57 | 84.00 | 2.15 | 5.07 | 70.89 | 71.17 | 71.21 | 70.88 | 0.10 | 4.27 | 57.9 |
| PSMGD | 0.12 | **0.25** | 77.2 | **74.4** | 3.01 | 6.61 | 103.0 | 103.5 | 103.7 | 101.6 | **0.09** | 5.64 | 92.4 |
| RobGrad | 0.11 | **0.25** | 82.7 | 80.3 | 2.31 | 5.30 | 70.6 | 70.3 | 71.1 | 69.1 | **0.09** | **2.64** | **57.7** |

The metric values in Table 8 for the baseline methods are taken from their original paper. Next, we provide RobGrad's result with error bars as follows, i.e., Table 9.

Table 9: The results on the QM-9 dataset (11 tasks). The mean and standard deviation (stdev) of each metric (↑ indicates higher better and ↓ indicates lower better) is reported over 3 independent runs.

| Method | $\mu$ | $\alpha$ | $\epsilon_{\text{HOMO}}$ | $\epsilon_{\text{LUMO}}$ | $\langle R^2 \rangle$ | ZPVE | $U_0$ | $U$ | $H$ | $G$ | $c_v$ | $\Delta M\% \downarrow$ |
|---|---|---|---|---|---|---|---|---|---|---|---|---|
| | | | | | | MAE ↓ | | | | | | |
| RobGrad (mean) | 0.11 | 0.25 | 82.7 | 80.3 | 2.31 | 5.30 | 70.6 | 70.3 | 71.1 | 69.1 | 0.09 | 57.7 |
| RobGrad (stdev) | ± 0.0023 | ± 0.0035 | ± 1.8 | ± 1.9 | ± 0.09 | ± 0.26 | ± 1.6 | ± 1.6 | ± 1.6 | ± 1.7 | ± 0.0011 | ± 3.24 |

# E    THE USE OF LARGE LANGUAGE MODELS (LLMS)

In this work, we used a large language model (LLM) as an assistive tool for language polishing. We clarify that while the LLM was involved in this way, all content ultimately attributed to the authors has been reviewed, verified, and edited by the authors. We take full responsibility for all parts of the manuscript. This usage of the LLM is disclosed here and in the submission form according to ICLR 2026 policies.

