# OpenReview forum: "RobGrad: Robustness-driven gradient descent for stochastic multi-objective optimization"
_ICLR.cc/2026/Conference — ICLR 2026 Conference Withdrawn Submission_

### Official Review · Reviewer_NuD6 · 2025-10-26

**Soundness:** 2
**Presentation:** 2
**Contribution:** 2
**Rating:** 2
**Confidence:** 5

**Summary:**

This paper proposes a robust stochastic multi-objective optimization (MOO) method for multi-task learning. The proposed method is built from Tchebysheff scalarization, a standard method in MOO, with an additional optimization term related to Pareto stationarity measure. Theoretical convergence rate of the proposed method is analyzed. Empirical studies on some benchmarks are presented.

**Strengths:**

1. The paper proposes a robust stochastic multi-objective optimization method for multi-task learning, which is an important problem in optimization and machine learning.

2. The paper provides convergence analysis and empirical studies on some benchmarks.

**Weaknesses:**

1. A major weakness is that the ```convergence guarantees are too weak```.
It requires bounded objective assumption, and the convergence rate in the nonconvex case (Thm 2, O(K^{-1/6})) is too slow. It is much worse than the existing results. Check [1,2,3] for example, and the survey paper [4]. Some existing methods do not require the bounded objective assumption, and have better convergence rates. Moreover, it is not even comparable to the convergence rate of linear scalarization.

2. ```There is no theoretical benefit of using the proposed method compared to existing stochastic MOO methods or even linear scalarization methods```. As mentioned by 1, only the convergence rate is slower with stronger assumptions. No other theoretical benefits are provided compared to existing stochastic MOO or scalarization methods. This leaves the audience wonder under what scenarios, the proposed method could be useful theoretically. More discussion is needed.



5. The notations and theoretical results are not presented clearly. For example, there is no $\xi$ in Theorems 1 or 2. See more in **Questions**.

6. Missing references. There are some other important works in stochastic multi-objective optimization, which are not discussed and compared in this paper. See a list below.

>[1] "Three-Way Trade-Off in Multi-Objective Learning: Optimization, Generalization and Conflict-Avoidance," L. Chen et al. NIPS 2023

>[2] "Variance reduction can improve trade-off in multi-objective learning," H. Fernando et al. ICASSP 2024

>[3] "Joint Gradient Balancing for Data Ordering in Finite-Sum Multi-Objective Optimization," H. Yang et al. ICLR 2025

>[4] "Gradient-based multi-objective deep learning: Algorithms, theories, applications, and beyond," W. Chen et al. arXiv 2025

**Questions:**

1. What is $\lambda^*$ in Theorem 1?

2. $x^*$ in Theorem 1 is one solution of (2), which is only a subset of $Y_{WN}$. The motivation for achieving this solution needs more explanation.

3. The motivation of this paper is not clear. The key idea is in Eq. (2). But when $\alpha = 0$, the method reduces to Tchebyshef scalarization, which yields weakly Pareto optimal or Pareto stationary solutions, with $\min_\lambda ||\nabla f(x)^\top \lambda ||=0$. What is the point of adding the additional minimum norm gradient term with parameter $-\alpha$? It seems to deviate from achieving Pareto stationarity.

4. Why do you provide convergence to the optimal solution of (2) in the convex case, but the convergence to Pareto stationarity in the nonconvex case? Can you discuss the stationary condition of (2) and convergence to it in the nonconvex case?

---

### Official Review · Reviewer_Xocf · 2025-10-27

**Soundness:** 2
**Presentation:** 3
**Contribution:** 2
**Rating:** 4
**Confidence:** 4

**Summary:**

This paper studies the stochastic multi-objective optimization problem and claims that recent methods are based on instantaneous gradients and lack a global optimization perspective, which may lead to suboptimal solutions. Thus, they propose a robustness-driven gradient descent (RobGrad) algorithm. Meanwhile, the theoretical analysis is provided.

**Strengths:**

1. The presentation of this paper is good and easy to follow.
2. The proof seems to be rigorous to me.
3. Sufficient illustration, Tables, and Figures help with understanding and comparison.

**Weaknesses:**

1. The authors point out that gradient-based MOO approaches lack a global optimization perspective and are sensitive to the initial decision in lines 83-85. I must admit that this is not straightforward to me, and could authors provide more explanation (either experimental results or analysis work)?
2. Besides, the method is claimed to improve the robustness of the worst-case of weight assignment and stability. The addition of the term $R(x)$ can indeed improve stability. But how does it improve the robustness?
3. I appreciate the idea of using "A PEDAGOGICAL EXAMPLE" before presenting the method; however, I do not believe this example motivates the RobGrad. If a balanced result is preferred, we can also add a constraint on the empirical loss and a regularization term so that $r(x)=l(x)$, following the paper https://arxiv.org/pdf/2502.08585.
3. In the algorithm, the weight is updated by optimizing problem (3). However, there is no description of this part. Though it is a quadratic problem, what is the cost of it?
4. In Theorems 1 and 2, the function values are bounded. Is this assumption necessary? SDMGrad does not have this assumption.
5. Meanwhile, Theorem 1 is in a "high probability" version. Why is there no "certain expectation upper bound" even with the function values' boundness?
6. [General question] The gradient-based method is not computationally efficient, with $p$ times the gradient computation, and recent scalarization methods are showing stronger performance (GO4Align, LDC-MTL). In this case, what are the benefits of RobGrad over them.

**Questions:**

Please check the weaknesses.

---

### Official Review · Reviewer_gcfw · 2025-10-31

**Soundness:** 2
**Presentation:** 2
**Contribution:** 2
**Rating:** 4
**Confidence:** 5

**Summary:**

This paper proposes RobGrad, a robustness-driven gradient descent method to address the stochastic multi-objective optimization problem. RobGrad leverages information from three aspects: $\mathcal{C}_1$ loss value balancing, $\mathcal{C}_2$ gradient conflict, and $\mathcal{C}_3$ weight smoothing. Theoretical analysis under both convex and non-convex settings is provided. Experiments demonstrate that the proposed method can achieve competitive performance compared to existing baselines.

**Strengths:**

1. The optimization formulation is clear. The objective leverages the loss, gradient conflict information, and the theoretical analysis justifies the proposed algorithm.
2. Extensive experiments are conducted to show the effectiveness of RobGrad.

**Weaknesses:**

1. Some parts of the paper need further clarification:
   - Lines 351-352: The citation of FairGrad is incorrect. The reported results are from [1], but were not cited correctly.
   - There may be typos in the proof of Lemma 1. The term $\alpha (\sum_{i=1}^p \lambda^i \nabla^2 f(x^\star)^\top \lambda^\star)$ seems should be $\alpha (\sum_{i=1}^p \lambda^i \nabla^2 f^i(x^\star)^\top)$. Please double-check.
   - Theorem 1 uses the surrogate $\mathcal{L}(x,\lambda)=\lambda^\top f(x)$, but the algorithm updates $\lambda$ with Eq. (3). Though Lemma 2-4 provide insights, it still reads inconsistently. Additional clarification or explanation would help improve coherence.
   - How to solve Eq. (3) should be made clear. It is a QP problem, and an exact solution can be obtained. Otherwise, if an approximated solution is used, then the theoretical analysis should account for the approximation error. Well, the attached code shows that an exact optimal solution is computed, but it would still be helpful to explicitly clarify this in the paper.

2. The proposed method lacks novelty. The loss value balancing term $\mathcal{C}_1$ is just a standard scalarization idea. The proximal term $\mathcal{C}_3$ is a common choice to stabilize the updates and make the inner problem strongly convex.

3. The analysis of non-convex setting adopts the bounded function value assumption, which is a stronger condition, but the convergence rate is the same, compared to existing baselines.

4. I am concerned that Eq. (3) is sensitive to loss scales. Consider different types of loss functions, such as cross-entropy, mIoU, mean absolute error (MAE), the resulting losses vary significantly in scale. Scaling task $i$ by​ $c_i$ (loss $l_i$ to $c_i l_i$) in Eq. (3), the value balancing term scales linearly in ​$c_i$, but the conflict term scales quadratically.


[1] Fair Resource Allocation in Multi-Task Learning. [ICML 2024]

**Questions:**

See the weaknesses discussed above.

---

### Official Review · Reviewer_vano · 2025-11-01

**Soundness:** 3
**Presentation:** 3
**Contribution:** 1
**Rating:** 2
**Confidence:** 5

**Summary:**

SMOO often struggles with gradient conflicts, and current methods focusing on instantaneous gradients can lead to suboptimal results. This paper reformulates the SMOO problem as a min-max optimization, aiming to minimize the worst-case objective value from a global perspective. The authors propose a new algorithm, RobGrad, which guarantees that all objectives perform reasonably well and demonstrates competitive or superior performance in multi-task learning experiments.

**Strengths:**

1. The paper organization is clear and easy to follow.
2. Provide a theoretical analysis for the proposed algorithm.
3. Provide comprehensive experimental results.

**Weaknesses:**

1. The motivation of the proposed method is unclear. The author claims that "these methods are based on instantaneous gradients and lack a global optimization perspective". What is "global optimization perspective"? I do not find a clear definition of this concept in this paper. Does the momentum update or the preference-based MOO method consider "global optimization"? Moreover, the logic from 3.2 to 4.1 is also confused. It seems section 3.2 does not provide any meaningful information. For the conducted experiments, such as NYUv2. The loss function of each task is completely different, so "all objectives are equally important" can not happen in most MTL settings. Then the authors directly proposed an optimization problem (3) and say that this optimization problem can solve the problem they focus on? Why? So compared with so many task balancing methods (CAGRAD, PCGRAD, MOCO, NASH-MTL, etc. ). Why does this method suddenly achieve SOTA in all benchmarks? Figure 3 tries to decompose each part in the optimization problem (3), each part is not a novel idea in the gradient-based MOO field.

2. No discussion on the theoretical result. No comparison with any previous work, such as assumptions and convergence rate. My intuitive understanding is that the proof in this paper is no more complex than that of SDMGrad, since both use iterative methods to solve for the weights, and both consider stochastic objectives.

3. Lacking a study on computational cost. About two or three years ago, I think similar approaches were advanced in solving MTL tasks. However, the current study shows that $\mathcal{O}(1)$ cost method is enough to achieve SOTA performance in the MTL problem, see FAMO and "Smooth Tchebycheff Scalarization for Multi-Objective Optimization". These approaches are fast and efficient and available for industrial tasks (such as training large-scale recommender systems). The proposed method needs to update $\lambda$ and save all gradients and update parameters, it is too heavy on both computational cost and memory cost.

**Questions:**

See the weaknesses part.

---

### Note · Authors · 2025-11-25

**Comment:**

Dear PCs, ACs, and Reviewers,

We would like to express our gratitude to the esteemed reviewers for providing valuable reviews, which help enhance the quality of our manuscript. After careful consideration, we have decided to withdraw the submission to further refine our work.

Thank you for your attention to this matter.

Sincerely, Authors of RobGrad.

**Withdrawal Confirmation:**

I have read and agree with the venue's withdrawal policy on behalf of myself and my co-authors.